# Enhancing Diversity in Bayesian Deep Learning via Hyperspherical Energy Minimization of CKA

**David Smerkous**
Oregon State University
Corvallis, OR, USA
smerkoud@oregonstate.edu

**Qinxun Bai**
Horizon Robotics
Sunnyvale, CA, USA
qinxun.bai@gmail.com

**Li Fuxin**
Oregon State University
Corvallis, OR, USA
lif@oregonstate.edu

## Abstract

Particle-based Bayesian deep learning often requires a similarity metric to compare two networks. However, naive similarity metrics lack permutation invariance and are inappropriate for comparing networks. Centered Kernel Alignment (CKA) on feature kernels has been proposed to compare deep networks but has not been used as an optimization objective in Bayesian deep learning. In this paper, we explore the use of CKA in Bayesian deep learning to generate diverse ensembles and hypernetworks that output a network posterior. Noting that CKA projects kernels onto a unit hypersphere and that directly optimizing the CKA objective leads to diminishing gradients when two networks are very similar. We propose adopting the approach of hyperspherical energy (HE) on top of CKA kernels to address this drawback and improve training stability. Additionally, by leveraging CKA-based feature kernels, we derive feature repulsive terms applied to synthetically generated outlier examples. Experiments on both diverse ensembles and hypernetworks show that our approach significantly outperforms baselines in terms of uncertainty quantification in both synthetic and realistic outlier detection tasks.

## 1 Introduction

Bayesian deep learning has always garnered substantial interest in the machine learning community. Instead of a point estimate which most deep learning algorithms obtain, a posterior distribution of trained models could significantly improve our understanding about prediction uncertainty and avoid overconfident predictions. Bayesian deep learning has potential applications in transfer learning, fairness, active learning, and even reinforcement learning, where reducing uncertainty can be used as a powerful intrinsic reward function (Yang & Loog, 2016; Ratzlaff et al., 2020; Wang et al., 2023).

One line of approach to Bayesian deep learning is to add noise to a single trained model. Such noises can either be injected during the training process, e.g. as in the stochastic gradient Langevin dynamics (Welling & Teh, 2011), or after the training process (Maddox et al., 2019). However, many such approaches often underperform the simple ensemble method (Lakshminarayanan et al., 2017b) which merely trains several deep networks with different random seeds. Intuitively, an ensemble, because it starts from different random initializations, might be able to "explore" a larger portion of the parameter space than those that are always nearby one specific model or training path. Because of this, ensembles may capture different modes and therefore better represent the posterior distribution of "well-trained" network functions (Fort et al., 2019; Wilson & Izmailov, 2020).

However, a critical question is, how different are the networks in an ensemble from one another? And can we utilize the idea of diversification to further improve these networks by making them even more diverse? In order to answer these questions, we first need a metric to compare those networks, which is in itself a significant problem; regular L1/L2 distances, either in the space of the network parameters, or in the space of the network activations, are not likely to work well. First, they suffer from the curse

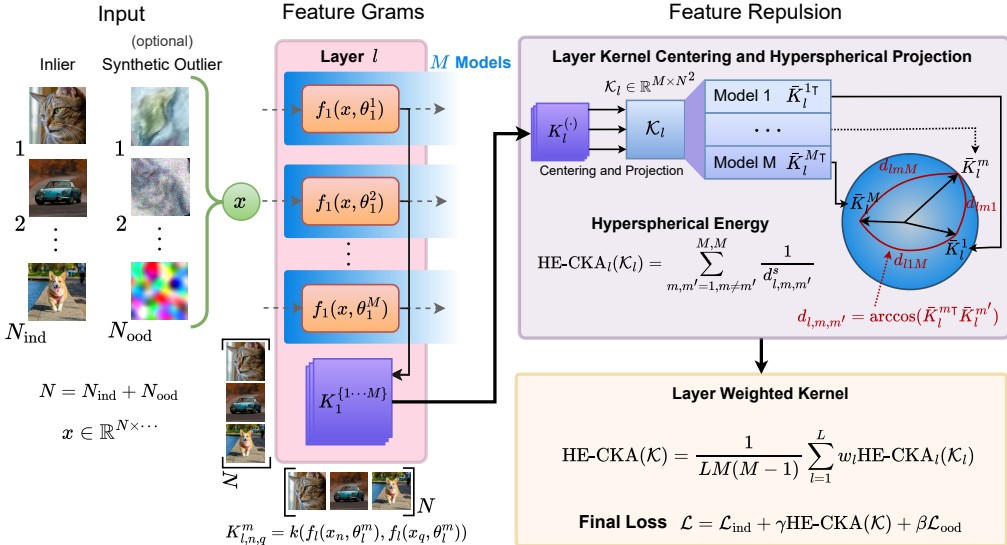

Figure 1: Overview of feature repulsive loss construction: Starting with a batch of examples (left), optionally including synthetic outliers, ensemble features at each layer $l$ are used to construct centered Gram matrices projected onto the unit hypersphere (middle). The hyperspherical energy is then calculated between models, weighted by layer, and incorporated into the loss function (right).

of dimensionality due to the excessive number of parameters in modern deep networks. Moreover, there is the peculiar *permutation invariance*, where one can randomly permute the different channels of each layer and result in a network that has vastly different parameters and activations, yet represents the same function. The popular RBF kernel lacks this permutation invariance inhibiting methods like Stein Variational Gradient Descent (SVGD) from working effectively on larger networks (D' Angelo & Fortuin, 2021). Therefore, a proper kernel for comparing network functions should address these critical issues by being effective in high-dimensional spaces and invariant to permutations of neural network channels.

Kornblith et al. (2019) proposed an interesting approach for performing this comparison based on Centered Kernel Alignment (CKA). The idea is, instead of directly comparing activations or parameters, comparison is made between the Gram matrices of the same dataset fed into two different networks. Each example will generate a feature vector at each layer of the network, and a kernel matrix can be constructed based on the similarity between all example pairs in the dataset. Then, a CKA metric measures the similarity of these two Gram matrices as the similarity of the two networks. This idea addresses the permutation invariance issue and generates meaningful comparisons between deep networks.

In this paper, we propose to **explicitly promote diversity** of network functions by adding CKA-based loss terms to deep ensemble learning. Given that CKA projects all kernels on a hypersphere, we further propose to use Hyperspherical Energy (HE) minimization as an approach to more evenly distribute the ensemble of neural networks on the hypersphere. Experiments on synthetic data, MNIST, CIFAR, and TinyImageNet show that our approach maintains the predictive accuracy of ensemble models while boosting their performance in uncertainty estimation across both synthetic and realistic datasets. Besides, we demonstrate that our method can also be applied to training hypernetworks, improving the diversity and uncertainty estimation of the networks generated by a hypernetwork. Additionally, we propose using synthetic out-of-distribution (OOD) examples, to reduce their likelihood, and introducing feature repulsive terms on synthetic outlier examples to enhance OOD detection performance. We hope that our approach provides a different perspective to variational inference methods and contributes to improving uncertainty estimation in deep networks. Code is publicly available at `https://github.com/Deep-Machine-Vision/he-cka-ensembles`.

## 2 Related Work

**Uncertainty Estimation.** A large body of literature has studied the problem of uncertainty estimation for neural networks. Bayesian neural networks (Gal & Ghahramani, 2016; Krueger et al., 2017;

Nazaret & Blei, 2022) approximate a posterior distribution over the model parameters and therefore estimate the epistemic uncertainty of the predictions. Non-Bayesian approaches, on the other hand, rely on bootstrap (Osband et al., 2016), ensemble (Lakshminarayanan et al., 2017b; Wen et al., 2020; Park & Kim, 2022), and conformal prediction (Bhatnagar et al., 2023) to generate multiple neural networks of the same structure. Our approach is most closely related to ensemble methods for estimating predictive uncertainty. We follow the common practice of evaluating uncertainty by distinguishing between inlier and outlier images for datasets like CIFAR/SVHN and MNIST/FMNIST. Most approaches typically evaluate the separation or distance within the feature space of a model between inliers and outliers (Mukhoti et al., 2023; Van Amersfoort et al., 2020; D' Angelo & Fortuin, 2021; Ovadia et al., 2019b; Lakshminarayanan et al., 2017a; Liu et al., 2020). Deep deterministic uncertainty (DDU) utilizes a single deterministic network with a Gaussian mixture model (GMM) fitted on the observed inlier features before the last layer and calculates separation of inliers and outliers using feature log density. We refer the reader to Mukhoti et al. (2023) for more information. For a more comprehensive survey and benchmarking of different uncertainty estimation approaches, refer to Ovadia et al. (2019a); Gawlikowski et al. (2022).

**ParVI.** Particle-based variational inference methods (ParVI), such as Stein Variational Gradient Descent (SVGD) (Liu & Wang, 2016; Chen et al., 2018; Liu & Zhu, 2018), use particles to approximate the Bayes posterior. Our work most closely resembles work done by D' Angelo & Fortuin (2021), which explores adapting kernelized repulsive terms in both the weight and function space of deep ensembles to increase model diversity and improve uncertainty estimation. Our work, however, focuses more on constructing a new kernel rather than exploring new repulsive terms that utilize an RBF kernel on weights or network activations.

**Hypernetworks.** Hypernetworks have been used for various specific tasks, some are conditioned on the input data to generate the target network weights, such as in image conditioning or restoration (Alaluf et al., 2022; Aharon & Ben-Artzi, 2023). It has seen popular use in meta-learning tasks related to reinforcement learning (Beck et al., 2023, 2024; Sarafian et al., 2021), and few-shot learning in Zhmoginov et al. (2022). Hypernetworks conditioned on a noise vector to approximate Bayesian inference have been proposed (Krueger et al., 2018; Ratzlaff & Fuxin, 2019), but either require an invertible hypernet or do not diversify target features explicitly. Our motivation is to provide Bayesian hypernetworks by explicitly promoting feature diversity in target networks.

## 3 Measurements of Network Diversity

In order to generate an ensemble of diverse networks, we first need a measurement of similarity between internal network features. Throughout this paper, we denote a deep network with $L$ layers as $f(x, \theta) = f_L(f_{(...)}(f_1(x, \theta_1), \cdots), \theta_L)$, where $f_l(x, \theta_l) \in \mathbb{R}^{p_l}$ are the features of layer $l$ parameterized by $\theta_l$, where $p_l \in \mathbb{N}$ is the output feature dimension.

### 3.1 Comparing two networks with CKA

We will compare two networks with the same architecture layer-by-layer. Given two networks at layer $l$ with weights $\theta_l^1, \theta_l^2$ and feature activations $f_l(x, \theta_l^1), f_l(x, \theta_l^2)$, a naive approach would be to take some Euclidean $L_k$ norm between the weights $\|\theta_l^1 - \theta_l^2\|_k$ or features $\|f_l(x, \theta_l^1) - f_l(x, \theta_l^2)\|_k$, but those tend to be bad heuristics for similarity measures in high-dimensional vector spaces due to the curse of dimensionality (Reddi et al., 2014; Aggarwal et al., 2001; Weber et al., 1998). A better approach to measuring similarity would be to analyze the statistical independence or alignment of features between networks, through Canonical Correlation Analysis (CCA), Singular Vector CCA (SVCCA), Projection-Weighted CCA (PWCCA), Orthogonal Procrustes (OP), Hilbert-Schmidt Independence Criterion (HSIC), or Centered Kernel Alignment (CKA)(Raghu et al., 2017; Gretton et al., 2005; Kornblith et al., 2019). Ideally, the chosen metric should be computationally efficient, invariant to isotropic scaling, orthogonal transformations, permutations, and be easily differentiable. However, CCA methods and OP require the use of Singular Value Decomposition (SVD) or iterative approximation methods, which can be computationally intensive. Additionally, HSIC and OP are not invariant to isotropic scaling of $f_l$.

As a comparison metric between networks, Kornblith et al. (2019) propose to utilize CKA on Gram matrices, obtained by evaluating the neural network on a finite sample. CKA is based on the non-parametric statistical independence criterion HSIC, which has been a popular method of measuring

statistical independence as a covariance operator in the kernel Hilbert spaces (Gretton et al., 2005). An empirical estimation of HSIC on a dataset of $N$ examples is given by $1/(N-1)^2 \mathrm{tr}(K^1 H K^2 H)$, where the two Gram matrices $K^1_{i,j} = k(f_l(x_i, \theta^1_l), f_l(x_j, \theta^1_l))$ and $K^2_{i,j} = l(f_l(x_i, \theta^2_l), f_l(x_j, \theta^1_l))$ are constructed through the $k(\cdot, \cdot)$ kernel function, and $H = I - \frac{1}{N}\mathbf{1}\mathbf{1}^\mathsf{T}$ a centering matrix to center the Gram matrices around the row means, where $\mathbf{1}$ denotes the all ones vector, and $I$ as the identity matrix. This function, however, is not invariant to isotropic scaling. The isotropic scaling invariant version of HSIC is termed Centered Kernel Alignment (CKA) (Kornblith et al., 2019),

$$\mathrm{CKA}(K^1, K^2) = \frac{\mathrm{HSIC}(K^1, K^2)}{\sqrt{\mathrm{HSIC}(K^1, K^1)\mathrm{HSIC}(K^2, K^2)}}. \tag{1}$$

We stick with the linear kernel for $k$, unless otherwise specified, due to its computational simplicity. The RBF kernel works, but it is computationally expensive and requires the use of heuristic like the median heuristic to perform well and make CKA isotropic scaling invariant (Reddi et al., 2014; Kornblith et al., 2019).

### 3.2 Generalizing to multiple networks

Given an ensemble of $M$ models, a simple approach to generalizing Eq. (1) to measure the similarity of an ensemble would be to construct a pairwise alignment metric. For each layer $l$ of each member of the ensemble $m$, we construct the set of kernel matrices $\mathcal{K} = \{K^m_l\}^{m=1,\ldots,M}_{l=1,\ldots,L}$. The mean pairwise loss across all layers $L$ is as follows,

$$\mathrm{CKA}_{\mathrm{pw}}(\mathcal{K}) = \frac{1}{LM(M-1)} \sum_{l=1}^{L} \sum_{\substack{m,m'=1 \\ m \neq m'}}^{M,M} \mathrm{CKA}(K^m_l, K^{m'}_l), \tag{2}$$

In its current form, $\mathrm{CKA}_{\mathrm{pw}}$ provides a good approximate metric to evaluate the similarity among members of an ensemble. We found that rewriting Eq. (2) gives us another perspective on optimizing CKA. First, to simplify notation, let $\bar{K}^m = \frac{1}{\|K^m H\|_F} \mathrm{vec}(K^m H)$ be the centered and normalized Gram matrix, and rewriting the inner product in Eq. (1) results in the cosine similarity metric $\mathrm{CKA}(K^m, K^{m'}) = \bar{K}^{m\top}\bar{K}^{m'}$. The matrix of the vectorized kernels from the set $\mathcal{K}_l$ can be represented in a compact form $\mathbf{K}_l$, and $\mathrm{CKA}_{\mathrm{pw}}$ can be rewritten using this compact form,

$$\mathrm{CKA}_{\mathrm{pw}}(\mathcal{K}) = \frac{1}{LM(M-1)} \sum_{l=1}^{L} \mathbf{1}^\mathsf{T} \mathrm{zd}(\mathbf{K}_l \mathbf{K}_l^\mathsf{T})\mathbf{1} \quad \text{s.t } \mathbf{K}_l = \begin{bmatrix} \bar{K}_l^{\ 1} \\ \vdots \\ \bar{K}_l^{\ M} \end{bmatrix} \in \mathbb{R}^{M \times N^2} \tag{3}$$

where $\mathrm{zd}(X) = X \odot (\mathbf{1}\mathbf{1}^\mathsf{T} - I)$ is a function that zeros out the diagonal of a matrix.

Now each row $m$ of $\mathbf{K}_l$ is a vectorized Gram matrix with unit length from the model $m$. We can view these vectors as the Gram matrices projected on the unit hypersphere as shown in Fig. 1. For each pair of models $i, j$ on the hypersphere, with an angle $\phi_{i,j}$ between the feature gram vectors, CKA is equivalent to $\cos(\phi_{i,j})$. Thus minimizing pairwise CKA would reduce the sum of $\cos(\phi_{i,j})$, pushing Gram matrices between model pairs apart.

### 3.3 Comparing Networks with Hyperspherical Energy

Note that CKA suffices as a differentiable measure between deep networks and one can directly minimize CKA to push different models in the ensemble apart from each other. However, CKA may have a specific deficiency as an optimization objective in that the gradient of $\cos(\phi)$ is $-\sin(\phi)$, which is close to 0 when $\phi$ is close to 0. In other words, if two models are already very similar to each other (their CKA being close to 1), then optimizing with CKA may not provide enough gradient to move them apart. Hence, we explore further techniques to alleviate this drawback. Minimum Hyperspherical Energy (MHE) (Liu et al., 2021) aims to distribute particles uniformly on a hypersphere, which maximizes their geodesic distances from each other. In physics, this is analogous to distributing electrons with a repellent Coloumb's force.

Inspired by MHE, we propose to adopt the idea of hyperspherical energy (HE) on top of the CKA kernel to compare neural networks, termed HE-CKA, which is novel to our knowledge. For each

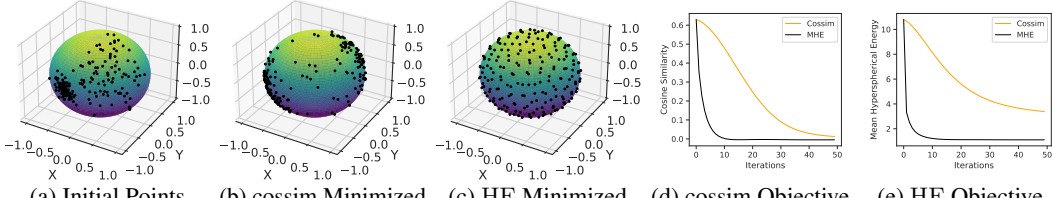

(a) Initial Points    (b) cossim Minimized    (c) HE Minimized    (d) cossim Objective    (e) HE Objective

Figure 2: Comparison between optimizing cosine similarity (cossim) or HE on a sphere. (a) initial random set of points placed on sphere. (b-c) the final set of points after 50 iterations either cossim or HE as the similarity metric. (d-e) the value of cossim/HE with respect to the number of iterations. The orange line indicates that cossim is minimized and the black line indicates that HE with $s = 2$ is minimized. Both methods used gradient descent with a learning rate of 0.75 and momentum 0.9.

layer $l$ we treat the $M$ model Gram vectors $\bar{K}_l^m$ as particles on the hypersphere, its geodesic on the hypersphere is then $d_{i,j} = \arccos(\text{CKA}(K_l^i, K_l^j)) = \arccos(\bar{K}_l^{i\intercal}\bar{K}_l^j)$, we define the energy function by simulating a repellent force on the particles via $F_{i,j} = (d_{i,j})^{-s}$ as shown in Fig 1. Incorporating this across all layers, weighted by $w_l$, and model pairs results in the overall hyperspherical energy of CKA between all models is.

$$\text{HE-CKA}(\mathcal{K}) = \frac{1}{LM(M-1)} \sum_{l=1}^{L} \sum_{\substack{m,m'=1 \\ m' \neq m}}^{M,M} \left( \arccos(\bar{K}_l^{m\intercal}\bar{K}_l^{m'}) \right)^{-s}, \tag{4}$$

where $s > 0$ is the Riesz $s$-kernel function parameter. For more information regarding the layer weighting $w_l$ and smoothing terms please see Appendix C.

HE has been shown as a proper kernel (Liu et al., 2021). The minimization of HE, as mentioned in Liu et al. (2021), asymptotically corresponds to the uniform distribution on the hypersphere, In order to demonstrate the difference between HE and the pairwise cosine similarity, we conducted a test on a synthetic dataset by generating random vectors from two Gaussian distributions in $\mathbb{R}^3$, and projecting on the unit hypersphere. We then minimized pairwise cosine similarity and HE respectively. Figure 2 illustrates that minimizing HE converges faster and achieves a more uniform distribution compared to minimizing cosine similarity. Specifically, as observed in Figure 2 (b), minimizing the cosine similarity loss caused particles to cluster towards two opposite sides of the sphere, as the gradient of this optimization – as mentioned in the beginning of the subsection, – becomes very small between particles that are clustered together. In Fig. 2(d), we show that minimizing HE actually leads to lower cosine similarity than directly minimizing cosine similarity, showing that minimizing cosine similarity could fall into local optima as described.

## 4    Particle-based Variational Inference by Minimizing Model Similarity

Armed with the comparison metrics between deep networks, we now proceed to incorporate the minimization of network similarity into deep ensemble training. In this section, we explore two different types of ensembles. The first is a regular ensemble where deep networks are trained to maximize the data likelihood, and we would add a term minimizing model similarity to it. Afterwards, we also explore the application of the idea on *generative ensembles* by hypernetworks, which aims to train a generator that generates network weights so that one can directly sample the posterior from it. Such a generator can easily exhibit mode collapse by always generating the same function, and we hope the idea of minimizing the similarity of generated networks would help alleviate this issue.

Suppose we are given a deep network with $L$ layers $f(x, \theta) = f_L(f_{(...)}(f_1(x, \theta_1), \cdots), \theta_L)$. We denote the ensemble of target network layer at layer $l$ as $E_l(x, \theta) = [f_l(x, \theta^1), ..., f_l(x, \theta^M)]$, with the ensemble parameters $\theta = \{\theta^m\}_{m=1}^{M}$, and the training set of $N$ examples as $\mathcal{D} = \{x_i, y_i\}_{i=1}^{N}$. From a Bayesian perspective, incorporating $\text{CKA}_{\text{pw}}$/HE-CKA into the ensemble training can be interpreted as imposing a Boltzmann prior with HE-CKA over the ensemble network parameters $\theta$ that produce feature Gram matrices uniformly distributed on the unit hypersphere. Specifically, $p(\theta) \propto \exp(-\gamma \text{ HE-CKA}(\mathcal{K}(E_{(\cdot)}(\cdot, \theta)))$, where $\mathcal{K}(E_{(\cdot)}(\cdot, \theta))$ is the set of feature Gram matrices, constructed from the ensemble $E$, as described in Sec. 3.2. The posterior distribution now becomes:

$$p(\theta|\mathcal{D}) \propto p(\mathcal{D}|\theta) \exp(-\gamma \text{ HE-CKA}(\mathcal{K}(E_{(\cdot)}(\cdot, \theta))) \tag{5}$$

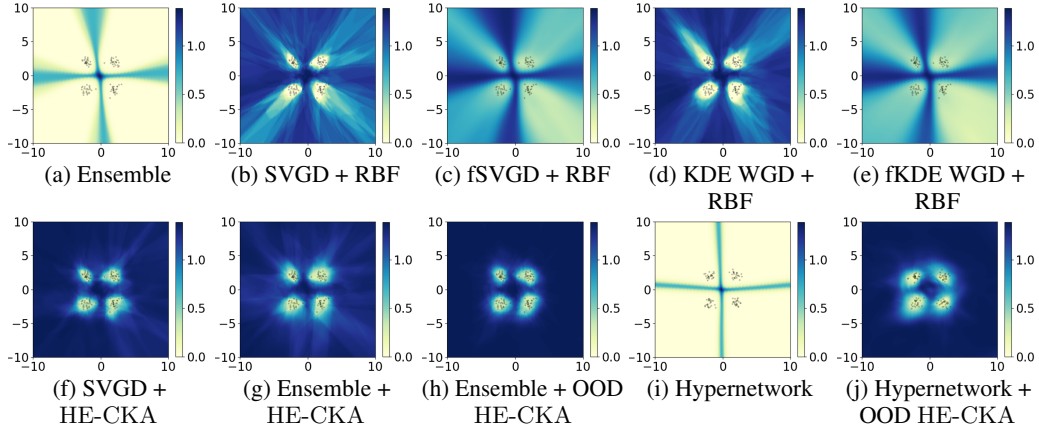

Figure 3: Predictive entropies (PE) on a four-cluster 2D classification task. Darker values indicate higher entropy, lower confidence regions, and lighter values indicate higher confidence regions. (b) and (d) use an RBF kernel on ensemble member weights, whereas (c) and (e) use an RBF kernel on ensemble member outputs. (f) and (g) use the HE-CKA, RBF feature kernel, for feature diversity on inlier points. Both (h) and (j) use HE-CKA and OOD entropy terms. All methods were trained on an ensemble of 30 four layer MLPs for 1k iterations with the same seeds.

The MAP estimate of the posterior in Eq. (5) results in the following objective:

$$\min_{\theta} M^{-1} \sum_{m=1}^{M} \left[ \sum_{i=1}^{N} \mathcal{L}(f(x_i, \theta^m), y_i) \right] + \gamma \, \text{HE-CKA}(\mathcal{K}(E_{(\cdot)}(\cdot, \theta)), \tag{6}$$

The left hand side is the negative log likelihood term where $\mathcal{L}(x, y)$ is the target loss, such as cross-entropy or MSE. Minimizing $\theta$, while adjusting the constant $\gamma$ used in the Boltzmann prior, allows us to balance between gram matrix hyperspherical uniformity and fitting the training data. Further explanation of Eq. (6)'s relationship to ParVI is given in Appendix A. Note that the same approach can be used to derive the formula for the CKA kernel in Eq. (2) as well.

### 4.1 Diverse Generative Ensemble with Hypernetworks

Besides diversifying ensemble models, we also explore using $\text{CKA}_{\text{pw}}$ / HE-CKA in learning a non-deterministic generator (Krueger et al., 2018) which gives us the ability to sample from a continuous nonlinear posterior distribution of network weights. This is appealing since it can generate any amount of network with a single training run of the generator, without being restricted by the fixed amount of posterior samples one can access with a regular ensemble.

The approach we take uses the concept of hypernetworks (Ha et al., 2016; Krueger et al., 2018). However, current variational inference methods are not scalable to larger models and generally require a change of variables or invertible functions (Krueger et al., 2018). Naively using a hypernetwork to transform a prior distribution to generate $\theta$ of the target network may result in the collapse of the posterior $\theta$ distribution. Hence, it would be interesting to explore using $\text{CKA}_{\text{pw}}$ / HE-CKA to avoid such mode collapses. We use the surrogate diversity loss in Eq. (4) to impose non-parametric independence of feature distributions. With hypernetworks we aim to transform, using a network $h(z)$, some prior distribution $z \sim \mathcal{N}(0, I), z \in \mathbb{R}^P$ to $h(z) = \theta \in \mathbb{R}^{\sum_l w_l}$, where $P$ is the dimensionality of the latent space, and $w_l$ the number of parameters for layer $l$. To learn the function $h(\cdot)$ we sample a batch $M$ of $\theta$'s, feed through the ensemble $E(x, \theta)$ and calculate loss, similar to a fixed ensemble as in Eq. (6). With the difference being that now we are backpropagating gradients to $h(\cdot)$ accumulated from the $M$ ensemble members.

Using a plain MLP for the hypernetwork $h$ would require the last layer's weight matrix to contain $\sum_l w_l \times J$ entries, where $J$ is the activation dimension right before the last layer. This could possibly result in a matrix of millions of trainable parameters. To overcome this challenge we follow the approach by Ratzlaff & Fuxin (2019) of decomposing $h$ into several parts. First a layer code generator $h(z) = \mathbf{c} \in \mathbb{R}^{L, c_{\text{size}}}$, and the layer generators $\theta_l = g_l(c_l)$, where each layer generator $g$ is a separate smaller network per layer $l$. See Fig. 4 for a visualization. Note $\mathbf{c}$ is a matrix with $L$ layer codes of size $c_{\text{size}}$.

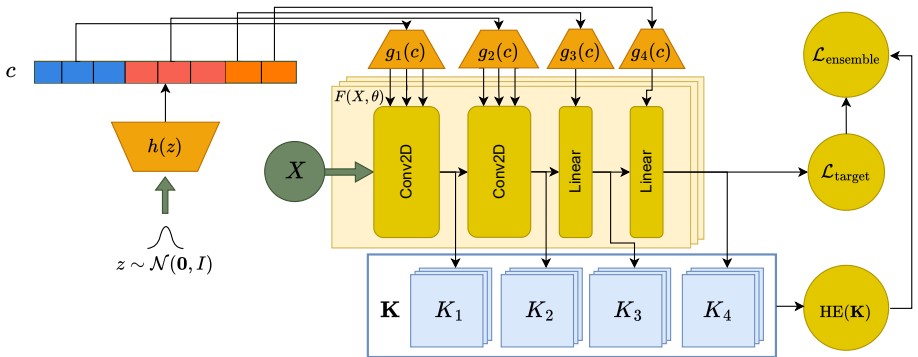

Figure 4: Hypernetwork $h(z)$ model architecture example on a four layer CNN

To further reduce size of the hypernetwork, for convolutional networks, we use the assumption that filters in convolutional layers can be independently sampled. For each convolutional layer $l$ we create layer code vectors via the layer code generator $c_l = h(z_l)$, where each code vector $i$ in $c_{l_i}$ corresponds to a latent vector for a single convolution filter $i$. We feed each filter code $i$ through a filter generator $g_l(c_{l_i})$ separately to generate the filter for layer $l$. An example architecture can be seen in Fig. 4.

## 4.2 Synthetic OOD Feature Diversity

Striking a balance between ensemble member diversity and inlier performance is a challenge. Enforcing strong feature dissimilarity on observed inlier examples could degrade inlier performance if not tuned correctly. ParVI methods that only observe inlier points, like SVGD, can achieve better diversity but often at the expense of inlier accuracy (D' Angelo & Fortuin, 2021). We have found that a more effective strategy is to reduce the feature similarity on obvious OOD examples, and reduce their likelihood, which could be synthetically generated. Intuitively, we want more diverse features on obvious outlier examples to indicate uncertainty because the networks trained on these examples should not be confident. We found this approach to generate OOD examples and increase their feature diversity to be very effective.

Importantly, the OOD points do not need to be close to the inlier data manifold at all. For images, we generate outlier points via random grids, lines, perlin noise, simplex noise, and vastly distorted and broken input samples. See Appendix E.2 for more details and example images. For vector datasets, such as the test 2D datasets presented in Fig. 3, we identify outlier points by locating the minimum and maximum values across training examples. Generally, the boundary does not need to be close to the in-distribution (ID) dataset to achieve good results. We split the Gram matrices into $K_{\text{ID}}$ and $K_{\text{OOD}}$ and apply HE-CKA to them separately, with respective hyperparameters $\gamma_{\text{ID}}$ and $\gamma_{\text{OOD}}$. The parameter value $\gamma_{\text{ID}}$ can be adjusted to be smaller than $\gamma_{\text{OOD}}$. Additionally, for classification tasks, we add an entropy-maximizing term, scaled by hyperparameter $\beta$, for synthetic OOD points to Eq. (6). Similar loss terms may be constructed for other tasks, such as variance for regression tasks, but we have not explored them yet.

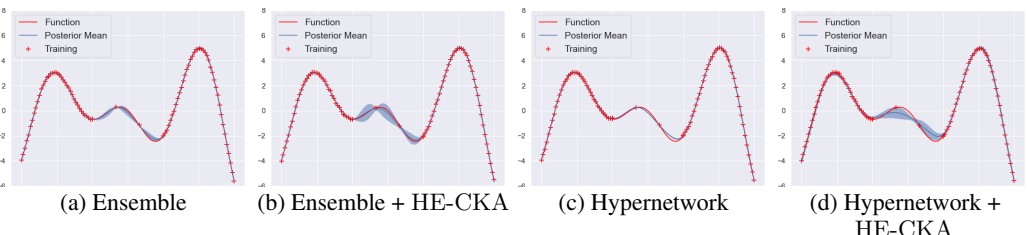

(a) Ensemble     (b) Ensemble + HE-CKA     (c) Hypernetwork     (d) Hypernetwork + HE-CKA

Figure 5: 1D regression task comparing uncertainty estimation between different approaches

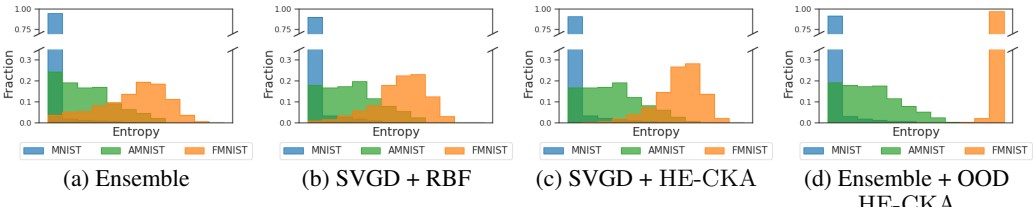

| (a) Ensemble | (b) SVGD + RBF | (c) SVGD + HE-CKA | (d) Ensemble + OOD HE-CKA |

Figure 6: Predictive softmax entropy between MNIST, Dirty-MNIST (with aleatoric uncertainty), and OOD Fashion-MNIST. Utilizing an ensemble of 5 LeNets. It can be seen that HE-CKA and OOD HE-CKA better separates the inlier Dirty-MNIST from outlier Fashion-MNIST.

## 5    Experiments

In this section, we conduct experiments on several datasets, ranging from synthetic tasks to realistic out-of-distribution (OOD) detection problems, to validate our approach. We compare OOD results between ensembles, ParVI, and other baselines.

### 5.1    Synthetic Data

We start by testing our approach on two synthetic tasks to visually assess the uncertainty estimation capability on both classification and regression problems. The first task is a 2D four-class classification problem, where each class is distributed in one quadrant of the 2D space with points sampled from Gaussians with $\sigma = (.4, .4)$ and $\mu = (\pm 2, \pm 2)$. The objective is to evaluate whether the models can accurately predict uncertainty, ideally showing low uncertainty near training examples and high uncertainty elsewhere. We employed a three-layer MLP trained with cross-entropy on the four classes and measured the predictive entropy of points sampled uniformly from a $10 \times 10$ grid.

When using cross-entropy alone, the decision boundaries among the four classes tend to be very similar. Deep ensembles classify with high confidence in most areas where they have never observed data before (Fig. 3(a)). Introducing the HE-CKA diversity term to the ensemble significantly reduces the ensemble's confidence on points outside the in-distribution set (Fig.3(g)). Furthermore, incorporating the HE-CKA and entropy term for OOD points allows the model to better estimate uncertainty, with only inliers being confident (Fig. 3(h)). In the case of hypernetworks, we observe the importance of a diversity term. Without it, hypernetwork predictions tend to be overconfident on outliers (Fig. 3(i)). However, when introducing HE-CKA hypernetwork, we achieve results closely resembling that of the ensemble + HE-CKA term (Fig. 3(j)).

In our second test, we perform a 1D regression modeling task. We aim to learn the function $y(x) = -\sin(1.2x)(1 + x)$ within $x \in (-6, 6)$ with high certainty everywhere except in $x \in (-2, 2)$. The training dataset involves sampling the function with 40 points uniformly from both $(-6, -2)$ and $(2, 6)$, with 2 points from $(-2, 2)$. We then fit a four layer MLP to approximate $y(x)$.

The visual result of each method is shown in Fig. 5. The fixed ensemble (Fig. 5(a)) has little diversity between the areas with low density, in contrast to the ensemble plus the HE-CKA term (Fig. 5(b)). The hypernetwork, without any feature diversity term (Fig. 5(c)) collapses, producing very similar weights. However, adding the HE-CKA term to the hypernetwork (Fig. 5(d)) alleviates this issue.

### 5.2    OOD Detection on Real Datasets

We evaluated our proposed approach on a variety of real-world datasets, including Dirty-MNIST, Fashion-MNIST, CIFAR-10/100, SVHN, and TinyImageNet. We employ different CNN architectures such as LeNet, ResNet32, and ResNet18 to demonstrate the versatility of our method across models of varying complexity. Our experiments compare the out-of-distribution (OOD) detection performance of our approach against several approaches, including Deep Deterministic Uncertainty (DDU), deep ensembles and Stein Variational Gradient Descent (SVGD) equipped with the RBF kernel.

We provide experimental settings and training details here and additionally in Appendix C. Limitations of this approach are discussed in Appendix D, while further insights into memory usage and computational efficiency are discussed in Appendix G. Details regarding synthetic OOD example generation is described in Appendix E.2.

Table 1: OOD detection results with inlier Dirty-MNIST and outlier Fashion MNIST, over 5 runs. All models were trained on a LeNet, with HE-CKA and $\text{CKA}_{\text{pw}}$ utilizing a cosine similarity feature kernel. One exception to predictive entropy (PE) report is DDU, which uses feature space density, indicated by a star, to calculate AUROC Mukhoti et al. (2023). More training details can be found in Appendix C.

| MODEL | NLL ($\downarrow$) | ACCURACY ($\uparrow$) | ECE ($\downarrow$) | AUROC FASHIONMNIST ($\uparrow$) | |
| --- | --- | --- | --- | --- | --- |
| | | | | PE | MI |
| DDU | $0.278 \pm 0.001$ | $82.177 \pm 0.032$ | $3.952 \pm 0.317$ | $94.168 \pm 3.425*$ | − |
| SINGLE | $\mathbf{0.272 \pm 0.002}$ | $82.299 \pm 0.166$ | $2.908 \pm 0.129$ | $65.935 \pm 10.669$ | $50.000 \pm 0.000$ |
| ENSEMBLE | $\mathbf{0.271 \pm 0.001}$ | $83.915 \pm 0.084$ | $\mathbf{1.306 \pm 0.098}$ | $86.095 \pm 1.608$ | $96.065 \pm 0.798$ |
| SVGD+RBF | $0.304 \pm 0.001$ | $83.560 \pm 0.072$ | $3.178 \pm 0.076$ | $91.003 \pm 1.155$ | $98.083 \pm 0.516$ |
| SVGD+$\text{CKA}_{\text{pw}}$ | $0.377 \pm 0.003$ | $82.351 \pm 0.150$ | $7.359 \pm 0.211$ | $89.195 \pm 4.260$ | $\mathbf{99.207 \pm 0.160}$ |
| SVGD+HE-CKA | $0.298 \pm 0.002$ | $83.879 \pm 0.110$ | $2.846 \pm 0.153$ | $94.380 \pm 1.332$ | $\mathbf{99.213 \pm 0.147}$ |
| HYPERNET | $0.278 \pm 0.003$ | $81.157 \pm 0.174$ | $4.253 \pm 0.092$ | $46.393 \pm 3.545$ | $64.856 \pm 2.768$ |
| HYPERNET+OOD HE-CKA | $0.325 \pm 0.014$ | $82.398 \pm 0.628$ | $3.058 \pm 0.665$ | $98.073 \pm 0.951$ | $77.548 \pm 9.526$ |
| ENSEMBLE+HE-CKA | $0.306 \pm 0.001$ | $83.684 \pm 0.029$ | $3.174 \pm 0.120$ | $94.656 \pm 1.095$ | $98.866 \pm 0.148$ |
| ENSEMBLE+OOD HE-CKA | $0.277 \pm 0.001$ | $\mathbf{84.090 \pm 0.049}$ | $1.712 \pm 0.061$ | $\mathbf{99.996 \pm 0.001}$ | $\mathbf{99.742 \pm 0.506}$ |

Table 2: OOD results on CIFAR10 vs SVHN. Methods used 10 particles of a modified ResNet32 were trained as described in D' Angelo & Fortuin (2021). Methods with ($*$) report values taken from D' Angelo & Fortuin (2021). One exception to predictive entropy (PE) report is DDU* which uses feature space density to calculate AUROC (Mukhoti et al., 2023).

| MODEL | NLL ($\downarrow$) | ACCURACY ($\uparrow$) | ECE ($\downarrow$) | AUROC SVHN ($\uparrow$) | |
| --- | --- | --- | --- | --- | --- |
| | | | | PE | MI |
| SVGD + RBF$_{(*)}$ | $0.287 \pm 0.001$ | $85.142 \pm 0.017$ | $5.200 \pm 0.100$ | $82.50 \pm 0.100$ | $71.00 \pm 0.200$ |
| FSVGD + RBF$_{(*)}$ | $0.292 \pm 0.001$ | $85.510 \pm 0.031$ | $4.900 \pm 0.100$ | $78.30 \pm 0.100$ | $71.20 \pm 0.100$ |
| KDE − WGD + RBF$_{(*)}$ | $0.276 \pm 0.001$ | $\mathbf{85.904 \pm 0.030}$ | $5.300 \pm 0.100$ | $83.80 \pm 0.100$ | $73.50 \pm 0.400$ |
| SGE − WGD + RBF$_{(*)}$ | $0.275 \pm 0.001$ | $85.792 \pm 0.035$ | $5.100 \pm 0.100$ | $83.70 \pm 0.100$ | $72.50 \pm 0.400$ |
| KDE − FWGD + RBF$_{(*)}$ | $0.282 \pm 0.001$ | $84.888 \pm 0.030$ | $4.400 \pm 0.100$ | $79.10 \pm 0.100$ | $75.80 \pm 0.200$ |
| SGE − FWGD + RBF$_{(*)}$ | $0.288 \pm 0.001$ | $84.766 \pm 0.060$ | $4.700 \pm 0.100$ | $79.50 \pm 0.100$ | $75.40 \pm 0.200$ |
| ENSEMBLE$_{(*)}$ | $0.277 \pm 0.001$ | $85.552 \pm 0.076$ | $4.900 \pm 0.100$ | $84.30 \pm 0.400$ | $73.60 \pm 0.500$ |
| SVGD + HE-CKA | $0.255 \pm 0.008$ | $\mathbf{85.890 \pm 0.381}$ | $3.675 \pm 0.089$ | $89.232 \pm 1.211$ | $70.977 \pm 1.445$ |
| SVGD + $\text{CKA}_{\text{pw}}$ | $0.286 \pm 0.002$ | $84.833 \pm 0.292$ | $4.945 \pm 0.185$ | $88.893 \pm 1.513$ | $70.327 \pm 1.763$ |
| DDU | $\mathbf{0.211 \pm 0.002}$ | $84.695 \pm 0.0361$ | $4.26 \pm 0.872$ | $86.281 \pm 0.020*$ | − |
| ENSEMBLE + OOD HE-CKA | $0.275 \pm 0.009$ | $\mathbf{86.133 \pm 0.482}$ | $5.436 \pm 0.599$ | $\mathbf{96.478 \pm 0.413}$ | $\mathbf{96.606 \pm 1.066}$ |
| HYPERNET + OOD HE-CKA | $0.259 \pm 0.002$ | $83.640 \pm 0.046$ | $\mathbf{1.115 \pm 0.050}$ | $88.121 \pm 0.182$ | $88.811 \pm 0.134$ |

**Dirty-MNIST vs Fashion MNIST.** The Dirty-MNIST vs Fashion MNIST OOD benchmark (Mukhoti et al., 2023) examines the capability of models to discern inliers, OOD data in a similar distribution and OOD data in a more dissimilar distribution. This dataset combines MNIST with more ambiguous and challenging examples known as ambiguous MNIST (AMNIST). We examine the ability to to distinguish MNIST and AMNIST from the out-of-distribution Fashion MNIST (FMNIST) (Xiao et al., 2017). We trained a LeNet5 on Dirty-MNIST and assessed OOD classification between Dirty-MNIST and FMNIST using predictive entropy (PE) and mutual information (MI). The area under the receiver operating characteristic (AUROC) is used to assess the separability between MNIST and FMNIST (Lecun et al., 1998; Bradley, 1997). For DDU a GMM is fit to the second to last layer's features over Dirty-MNIST, using the log density of features to distinguish inliers from outliers, rather than predictive entropy.

Results shown in Fig. 6 and Table 1. Ensembles (Fig. 6(a)) do not exhibit significant separation using predictive entropy (PE) alone, resulting in low AUROC with PE for FashionMNIST. While methods like SVGD, equipped with an RBF kernel, improve separation (Fig. 6(b)), our approach demonstrates that using HE-CKA on an ensemble alone surpasses both RBF kernels and DDU's approach, and when paired with OOD examples, and OOD likelihood minimization, results in almost perfect separation with 99.99% AUROC. CKA plots of each method are presented in Appendix F.

Table 3: OOD results on CIFAR-10 vs SVHN. Methods used a ResNet18 ensemble of size 5. ($\cdot$) indicates ensemble size.

| MODEL | OOD METHOD | NLL ($\downarrow$) | ACCURACY ($\uparrow$) | ECE ($\downarrow$) | AUROC PE SVHN ($\uparrow$) |
| --- | --- | --- | --- | --- | --- |
| WIDERESNET28-10+SN$_{Mukhoti\ et\ al.\ (2023)}$ (1) | GMM | − | 95.97 | 0.85 | 97.86 |
| RESNET18 (5) | PE | 0.122 | $\mathbf{96.34}$ | 1.08 | 96.18 |
| RESNET18 SVGD+RBF (5) | PE | 0.143 | 95.71 | 1.19 | 95.37 |
| RESNET18 SVGD+CKA (5) | PE | 0.125 | 96.25 | $\mathbf{0.41}$ | 96.07 |
| RESNET18 SVGD+HE (5) | PE | 0.124 | 96.23 | 0.63 | 96.01 |
| RESNET18+HE-CKA (5) | PE | $\mathbf{0.120}$ | 96.23 | 0.59 | 96.71 |
| RESNET18+OOD HE-CKA (5) | PE | 0.123 | 96.24 | 0.58 | $\mathbf{99.86}$ |

Table 4: Performance of a five member ResNet18 ensemble trained on TinyImageNet. All models utilized a pretrained deep ensemble with no repulsive term, then fine tuned for 30 epochs for each method (including deep ensemble). Methods utilizing $CKA_{pw}$ and HE-CKA utilized a linear feature kernel.

| MODEL | NLL ($\downarrow$) | ID ACCURACY ($\uparrow$) | ECE ($\downarrow$) | AUROC PE ($\uparrow$) | | |
|---|---|---|---|---|---|---|
| | | | | SVHN | CIFAR 10/100 | TEXTURES (DTD) |
| ENSEMBLE | 0.775 | 62.95 | 8.90 | 89.81 | 66.85/67.33 | 68.96 |
| SVGD+RBF | 0.926 | 61.87 | 16.10 | 92.76 | 72.23/73.73 | 65.67 |
| SVGD+CKA$_{pw}$ | 0.835 | 60.15 | 8.26 | 94.08 | 78.40/79.48 | 66.48 |
| SVGD+HE-CKA | **0.732** | 61.36 | **3.71** | 94.10 | 72.05/72.86 | 70.75 |
| ENSEMBLE+HE-CKA | 0.784 | **63.10** | 9.82 | 92.65 | 72.13/71.68 | 70.69 |
| ENSEMBLE+OOD HE-CKA | 0.786 | 61.88 | 8.02 | **99.31** | **81.56/87.64** | **90.94** |

**CIFAR-10/100 vs SVHN.** We further evaluated our method on CIFAR-10 and CIFAR-100 datasets, testing outlier detection performance on SVHN (Table 2) (Netzer et al., 2011). For a fair comparison with D' Angelo & Fortuin (2021), we trained ResNet32 ensembles following the training procedure and parameters described by D' Angelo & Fortuin (2021). For more details regarding model architecture please refer to the aforementioned paper and published code. We used predictive entropy and mutual information for the OOD classification, with the exception of DDU using feature space density (Mukhoti et al., 2023).

Given that the network presented in D' Angelo & Fortuin (2021) has significantly fewer parameters than a typical ResNet, it is expected to see an inferior classification accuracy to that of standard ResNet. In order to show that our approach generalizes to larger networks, we trained on larger ResNet18 ensembles. Results in Table. 3 show that HE-CKA can maintain similar accuracy as regular deep ensembles while significantly improving on ECE and AUROC of outliers. For the CIFAR-100 results please see Appendix C.3. Our ensemble with a standard ResNet18 with batch normalization even slightly outperforms a WideResNet-28-10 (WRN) using the approach by Mukhoti et al. (2023). Additionally, the mean inference time for a WRN is 13ms compared to 9ms for the ResNet18 ensemble on a Quadro RTX 8000.

**TinyImageNet vs SVHN/CIFAR-10/CIFAR-100/DTD.** To further evaluate the effectiveness of our approach to larger models and more complex datasets, we conducted experiments using the TinyImageNet dataset (Le & Yang, 2015). We trained ensembles of ResNet18 models and tested their ability to detect OOD samples from SVHN (Netzer et al., 2011), CIFAR-10/100 (Krizhevsky, 2009), and the Describable Textures Dataset (DTD) (Cimpoi et al., 2014). Our objective was to assess whether the proposed methods could generalize to large-scale settings and improve OOD detection performance without compromising in-distribution accuracy. Training details, and data splits, are provided in Appendix C.4.

Our proposed methods, especially Ensemble+OOD HE-CKA, enhanced OOD detection performance. Notably, Ensemble+OOD HE-CKA achieved an AUROC of 99.31% on SVHN and substantial improvements on CIFAR-10/100 and DTD datasets (Table. 4), with AUROC scores of 81.56%/87.64% and 90.94%, respectively. This improvement in OOD detection did not come at a major expense of ID accuracy.

## 6 Conclusion

In this paper, we explored the novel usage of CKA and MHE on feature kernels to diversify deep networks. We demonstrated that HE-CKA is an effective way to minimize pairwise cosine similarity, thereby enhancing feature diversity in ensembles and hypernetworks when applied on top of CKA. Our approach significantly improves the uncertainty estimation capabilities of both deep ensembles and hypernetworks, as evidenced by experiments on synthetic classification/regression tasks and real image outlier detection tasks. We showed that diverse ensembles utilizing predictive entropy alone can outperform other feature space density approaches, while synthetically generated OOD examples, far from the inlier distribution, can further significantly improve the OOD detection performance. While our current method requires fine-tuning several hyperparameters, such as layer weighting, we believe that future work could explore strategies for automatically estimating these parameters. We hope that our method inspires further advancements in Bayesian deep learning, extending its application to a wider range of tasks that require robust uncertainty estimation.

**Acknowledgements**

This work was funded in part by ONR award N0014-21-1-2052, DARPA HR001120C2022, NSF 1751412 and 1927564.

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

## A    Minimizing Model Similarity as Particle-based Variational Inference

In Bayesian deep learning, each network in an ensemble can be seen as a particle sampled from a distribution. Hence, the training process can be seen as a variational inference problem in terms of minimizing the KL-divergence between the empirical distribution defined by the particles and the training data. In this section, we relate Eq. (4) with an RKHS and apply it under a particle-based variational inference framework for supervised learning. Particle-based variational inference methods (Liu & Wang, 2016; Chen et al., 2018; Liu et al., 2019) can be viewed from a geometric perspective as approximating the gradient flow line on the Wasserstein space $\mathcal{P}_2(\mathcal{X})$ (Liu & Zhu, 2022). Let $q_t$ denote the gradient flow line of the KL-divergence w.r.t. some target (data) distribution $p \in \mathcal{P}_2(\mathcal{X})$, for an absolutely continuous curve $q_t$, its tangent vector at each $t$ is given by (Villani et al., 2009; Ambrosio et al., 2008),

$$\text{grad } \text{KL}(q_t|p) = -\nabla \log p + \nabla \log q_t. \tag{7}$$

The core idea of particle-based VI is to represent $q_t$ by a set of particles $\{x_i\}$ and adopt a first-order approximation of $q_{t+\epsilon}$ through a perturbation of $\{x_i\}$. In Eq. (7), while the first term corresponds to the maximum (data) likelihood term of supervised learning, the second term is intractable. Different variants of particle-based VI methods tackle this term via different approximation/smoothing methods. Inspired by SVGD (Liu & Wang, 2016), we approximate $\nabla \log q_t$ in an RKHS corresponding to the HE-CKA kernel.

In particular, let $\mathcal{T} = \{\phi : \mathcal{X} \to \mathcal{X}\}$ denote the space of transformations on space $\mathcal{X}$ of particles, a direction of perturbation can be viewed as a vector field on $\mathcal{X}$, which is a tangent vector in the tangent space $T_{\phi=\text{id}}\mathcal{T}$, where id is the identity transformation. Under the particle representation $\{x_i\} \sim q$, let $q_\phi(x)$ denote the distribution represented by transformed particles $\{\phi(x_i)\}$. To approximate $\nabla \log q$, we want to find the perturbation direction of $\{x_i\}$ that corresponds to the steepest ascend direction of the loss $\mathcal{J}(\phi) = \mathbb{E}_{x \sim q}[\log q_\phi(x)]$ at $\phi = \text{id}$, which is the gradient of $\mathcal{J}(\phi)$ in the tangent space $T_{\phi=\text{id}}\mathcal{T}$. This gradient is given by the following Lemma, with the proof given in Appendix B.

**Lemma A.1.** *For* $\mathcal{J}(\phi) = \mathbb{E}_{x \sim q}[\log q_\phi(x)]$,

$$\nabla_\phi \mathcal{J}(\phi)(\cdot)\Big|_{\phi=id} = \mathbb{E}_{x \sim q}\left[\nabla_x K_{\text{HE-CKA}}(x, \cdot)\right], \tag{8}$$

where $K_{\text{HE-CKA}}$ is the HE-CKA kernel defined by Eq. (4). In practice, Eq. 8 can be approximated by the empirical expectation,

$$\nabla_\phi \mathcal{J}(\phi)(\cdot)\Big|_{\phi=\text{id}} \approx \hat{\mathbb{E}}_{x \sim \{x_i\}}\left[\nabla_x K_{\text{HE-CKA}}(x, \cdot)\right]. \tag{9}$$

In this paper, we apply Eq. (7) and Eq. (9) to the supervised tasks of classification and regression.

## B    Proof of Lemma A.1

*Proof.* To compute the gradient of $\mathcal{J}(\phi) = \mathbb{E}_{x \sim q}[\log q_\phi(x)]$ at $\phi = \text{id}$, by definition, we compute as follows the differential of $\mathcal{J}$ at $\phi$, $\forall v \in T_{\phi=\text{id}}\mathcal{T}$,

$$d\mathcal{J}_\phi(v)\big|_{\phi=\text{id}} = \frac{d}{dt}\Big|_{t=0} \mathcal{J}(\phi + tv)\big|_{\phi=\text{id}}$$

$$= \mathbb{E}_{x \sim q}\left[\frac{d}{dt}\Big|_{t=0} \log q_{\phi+tv}(x)\right]$$

$$= \mathbb{E}_{x \sim q}\left[\frac{d}{dt}\Big|_{t=0} \log \frac{q(x)}{\left|\det\left(\frac{\partial(\phi+tv)}{\partial x}\right)\right|}\right]$$

$$= \mathbb{E}_{x \sim q}\left[\text{Tr}\left(\left(\frac{\partial \phi}{\partial x}\right)^{-1} \frac{d}{dt}\Big|_{t=0} \frac{\partial(\phi+tv)}{\partial x}\right)\Big|_{\phi=\text{id}}\right]$$

$$= \mathbb{E}_{x \sim q}\left[\sum_j \frac{\partial v^j}{\partial x^j}\right]$$

$$= \langle \mathbb{E}_{x \sim q}\left[\nabla_x K_{\text{HE-CKA}}(x, \cdot)\right], v\rangle_{\mathcal{H}}, \tag{10}$$

where $\mathcal{H}$ is the RKHS with corresponding HE kernel $K_{\text{HE-CKA}}$, and the following identities are used,

$$q_\phi(x) = \frac{q(x)}{\left| \det \left( \frac{\partial \phi}{\partial x} \right) \right|},$$

$$d \log |\det A| = \text{Tr}(A^{-1} dA),$$

$$\frac{\partial v^i(x)}{\partial x^j} = \langle \nabla_{x^j} K_{\text{HE-CKA}}(x, \cdot), v^i(x) \rangle_{\mathcal{H}}.$$

By definition of gradient,

$$d\mathcal{J}_\phi(v)\big|_{\phi=\text{id}} = \langle \nabla_\phi \mathcal{J}\big|_{\phi=\text{id}}, v \rangle_{\mathcal{H}},$$

comparing with Eq. 10,

$$\nabla_\phi \mathcal{J}(\phi)(\cdot)\big|_{\phi=\text{id}} = \mathbb{E}_{x \sim q} \left[ \nabla_x K_{\text{HE-CKA}}(x, \cdot) \right].$$

$\square$

## C  Training Details

### C.1  Smoothing Terms and Layer Weighting

To effectively train with the HE-CKA kernel for the repulsive term we found that it is essential to smooth out the particle energy using an $\epsilon_{\text{dist}}$ on the geodesics and $\epsilon_{\text{arc}}$ on the cosine similarity values. With larger smoothing terms we can reduce the large gradients on very similar particles, with CKA values near 1, and ensure other particles still receive some repulsive force. Additionally, Eq. (4) equally weighs every layer in the network. It has been empirically shown that the first few layers of deep neural networks have high similarity (Kornblith et al., 2019), which indicates that initial layers learn more aligned features. Enforcing strong hyperspherical uniformity, or low CKA, of feature Gram matrices may remove useful features. We have noticed that it is difficult to train models with a uniform HE-CKA layer weighting scheme of $1/L$. To fix this we applied a custom weighting scheme $w$ that typically increases linearly with the number of layers, with latter layers weighted higher. We found that using a weighting scheme in Eq. 4 allowed for finer control of the repulsive term. Typically the first layer in a CNN is a simple feature extractor, and depending on the depth of the network could assign too high of a repulsive term on the first layer. Additionally, the last layer could have too high of a weight and ruin inlier performance. We utilize a custom weighting scheme using the vector $\boldsymbol{w} = \{w_1, \cdots, w_L\}$, where $\|\mathbf{w}\|_1$ is typically 1. We define the smoothed HE-CKA version for training as $\text{HE}_{\text{smooth}}$ (Eq. 11).

$$\text{HE}_{\text{smooth}}(\mathbf{K}) = \frac{1}{M(M-1)} \sum_{l=1}^{L} w_l \sum_{\substack{m,m'=1 \\ m' \neq m}}^{M,M} \frac{1.0 + \epsilon_{\text{dist}}}{\arccos(\bar{K}_l^{m\intercal} \bar{K}_l^{m'} / (1.0 + \epsilon_{\text{arc}}))^s + \epsilon_{\text{dist}}} \tag{11}$$

The smoothing terms gives us finer control over the interaction between particles and prevents exploding or vanishing from the energy term. Although both $\epsilon_{\text{dist}}$ and $\epsilon_{\text{arc}}$ have a similar effect it is more important to include the $\epsilon_{\text{arc}}$ as the gradient of $\arccos$ approaches $\pm\infty$ near $-1$ and $1$ without any smoothing term. As demonstrated with the cosine similarity feature kernel used in Fig. 7. It is advised to set $\epsilon_{\text{dist}}$ as a small constant then vary $\epsilon_{\text{arc}}$ and $\gamma$ parameters when searching for the right kernel. Optionally, one may replace the Riesz-$s$ based kernel with an exponential one, ie $e^{-s \arccos(\bar{K}_l^{m\intercal} \bar{K}_l^{m'} / (1.0+\epsilon_{\text{arc}}))-\epsilon_{\text{dist}}}$, which provides a more numerically stable gradient, and more intuitive to understand growth term $s$. As discussed in Appendix. D the parameters for $\gamma$, $\beta$, and $w$ need to be selected. For MNIST experiments we performed a bayes sweep across parameters to select the layer weighting schemes, smoothing terms, and repulsive terms. For larger models, such as the CIFAR and TinyImageNet experiments, we selected, by trying a few combinations, a weighting schemes by testing values $\gamma \in [0.25, 1.5]$, $\beta \in [0.01, 10.0]$, and using layer weighting scheme $w_l$ that increases proportionally with $l$, and with different first layer and last layer values.

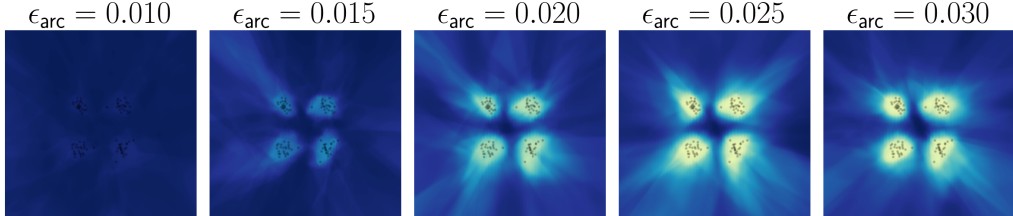

$\epsilon_{arc} = 0.010$   $\epsilon_{arc} = 0.015$   $\epsilon_{arc} = 0.020$   $\epsilon_{arc} = 0.025$   $\epsilon_{arc} = 0.030$

Figure 7: Effect of smoothing term when using a cosine similarity based HE-CKA$_{smooth}$ kernel with SVGD on inlier points only. All methods were trained with AdamW (lr=0.05, wd=0.0075), HE-CKA$_{smooth}$ $s = 2$, and $\epsilon_{dist} = 0.00025$, and $w = [0.2, 0.35, 0.85, 0.05]$ for 1k steps.

### C.2 Dirty-MNIST

The Dirty-MNIST experiments utilized an ensemble of 5 LeNet5 models with a modified variance preserving gelu activation function. Models were trained using AdamW with lr $= 0.0065$ and weight decay of $0.001$ for 50 epochs, except for Hypernetwork training which was trained for 85 epochs with AdamW with lr $= 0.0025$ and weight decay $0.0025$. Details such as lr warmup, gradient clipping, repulsive terms, layer weighting, HE-CKA smoothing terms, and more can be found in the official repository.

### C.3 CIFAR-10/100

Table 5: OOD results on CIFAR-100 vs SVHN. Methods used a ResNet18 ensemble of size 5.

| MODEL | OOD METHOD | NLL | ACCURACY ($\uparrow$) | ECE ($\downarrow$) | AUROC SVHN ($\uparrow$) |
|---|---|---|---|---|---|
| ENSEMBLE | PE | **0.74** | **81.81** | 5.77 | 89.62 |
| ENSEMBLE+HE-CKA | PE | **0.74** | 80.72 | **3.90** | 91.17 |
| ENSEMBLE+OOD HE-CKA | PE | 0.76 | 80.61 | 4.11 | **99.44** |

Additionally, we have some results showing much improvement on CIFAR-100 OOD detection with SVHN when trained with synthetic OOD examples in Table 5. With about a 10% improvement in AUROC between the inlier and outlier sets. We applied HE-CKA to an ensemble of ResNet18 models and evaluated the approach on CIFAR-10 (Table 3) and CIFAR-100 (Table 5). The models were trained for 200 epochs using SGD with a learning rate of $0.1$ and weight decay $5e$-$4$. The HE-CKA kernel used a linear kernel for feature calculation with the exponential kernel $s = 2$, and $\gamma = 1.0$. For experiments with out-of-distribution (OOD) data, the following values were adjusted: $\gamma = 0.5$, $\gamma_{OOD} = 0.75$, and $\beta = 0.75$. Details regarding layer weighting and smoothing are available in the repository. Forty-eight OOD samples were taken per batch for all CIFAR experiments, where applicable.

The feature repulsion term was not applied to every convolution of the ResNet18 architecture. To conserve computational resources, only a subset of layers was included. Specifically, the selected layers comprised the initial convolutional layer, the output of every other ResNet block within the first two of the four layers, the output of all blocks in the last two layers, and the final linear layer.

Training details regarding the ResNet32 experiments follow the training procedure, learning rate scheduling, and hyperparameters given by D' Angelo & Fortuin (2021). The hypernetwork variant, due to the difficulty of training, was trained for 180 instead of 143 epochs, and utilized group based normalization to stabilize feature variance.

### C.4 TinyImageNet and Particle Number Ablation

All models utilized a pretrained deep ensemble without any repulsive term and then fine-tuned using different methods, including our proposed approach. Methods utilizing CKA$_{pw}$ and HE-CKA employed a linear feature kernel. For OOD detection, we used predictive entropy (PE) computed from the ensemble predictions. Additionally, we generated synthetic OOD data from noise and augmented TinyImageNet samples to enhance the OOD detection capability. We utilized a training split of

80:10:10 for training, validation, and testing respectively. Training utilized SGD with a learning rate of $0.005$ and weight decay of $5e$-$4$. We additionally performed a particle number ablation on the ResNet18 + HE-CKA ensemble, utilizing the same repulsive term, showing improvements in accuracy, and outlier detection, when going from 2 particles to 5 (Table. 6).

Table 6: Ablation on number of training particles for ResNet18 + trained with HE-CKA TinyImageNet.

| PARTICLES | NLL (↓) | ID ACCURACY (↑) | ECE (↓) | AUROC PE (↑) | | |
|---|---|---|---|---|---|---|
| | | | | SVHN | CIFAR-10/CIFAR-100 | TEXTURES (DTD) |
| 2 | 0.791 | 59.21 | **5.21** | 89.87 | 67.34 | 68.48 |
| 3 | 0.771 | 60.86 | 5.74 | 91.57 | 69.05/70.59 | 69.31 |
| 4 | **0.761** | 62.26 | 6.90 | **92.92** | 70.83/71.44 | **70.89** |
| 5 | 0.784 | **63.10** | 9.82 | 92.65 | **72.13/71.68** | 70.69 |

# D    Limitations

With our approach, we are able to resolve some of the issues related to tackling permutation of feature channels, which normally pose challenges for Euclidean-based kernels like RBF. However, constructing a model kernel based on layer features requires tuning the repulsive term ($\gamma$), the likelihood term ($\beta$), and the layer weighting terms ($w$). This introduces numerous hyperparameters that need to be adjusted depending on the dataset and the architecture in use. Future work could explore automating the estimation of these parameters or simplifying the HE-CKA kernel. Although the assumption that the first few layers should have small repulsive terms seems clear, the weighting and smoothing of later layers remain unclear. This work only explored repulsive terms that increased with layer depth; the dynamics of which layers should have more repulsion are not well understood and have not yet been explored. Additionally, feature-based kernels based on $CKA_{pw}$ are sensitive to the number of particles and the batch size sampled, as the dimensionality of the hypersphere changes, impacting the repulsive terms. One possible solution could be to construct a normalized HE-CKA variant, which precomputes the minimum and maximum energy available on the $(N^2 - 1)$-sphere with $M$ models.

# E    Synthetic OOD examples

## E.1    OOD points for 2D datasets

The selection of an OOD set can help force ensemble members to learn unique internal features for outliers, resulting in more diverse output features. For simple datasets it can be easy to construct an OOD set by simply taking min/max + padding as shown in Fig. 8 for the 2D classification tasks.

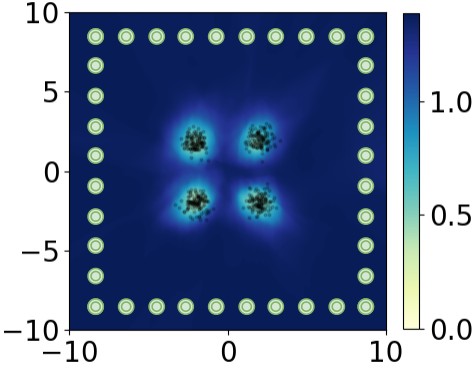

Figure 8: Example boundary set for 2d classification task. Note the boundary OOD set can be far away from in distribution points.

Table 7: Average pairwise unbiased CKA across all layers of an ensemble of 5 ResNet18 trained on CIFAR-10.

| METHOD | LAYER MEAN CKA |
|---|---|
| RESNET18 DEEP ENSEMBLE | 0.955 |
| RESNET18 SVGD+RBF | 0.972 |
| RESNET18 SVGD+CKA | 0.870 |
| RESNET18 SVGD+HE | 0.816 |
| RESNET18 HE | **0.479** |

## E.2 OOD Images for MNIST and CIFAR

Images are harder to define boundary/ood points, but we found in practice that generating images by transforming inliers to outliers via typical augmentations, and generating synthetic random channel data worked well in practice. Our approach to transforming inlier points to outlier points consists of the following augmentations in random order: blurring, affine transform, perspective transform, elastic deformations, erasures, Gaussian noise, Gaussian blurring and inversions. Examples of such images are shown in Fig. 9 for MNIST, Fig. 10 for CIFAR, and Fig. 11 for TinyImageNet. The ID to OOD set of images accounts for roughly 30-40% of the dataset, whereas the other 60% are randomly generated. Synthetic OOD images are generated via combinations of perlin noise, simplex noise, gaussian noise, lines, alternating grids, inversions, and random area thresholding. For images with more than one channel, such as CIFAR and TinyImageNet, we either apply different noise to each channel, use the same method but different seed, or occasionally broadcast one channel along all channels.

## F  MNIST CKA Results

We present the CKA plots of each method: Ensemble (Fig. 12), Hypernetwork (Fig. 13), SVGD + RBF (Fig. 14), SVGD + $\text{CKA}_{\text{pw}}$ (Fig. 15), SVGD + HE-CKA (Fig. 16), Ensemble + HE-CKA (Fig. 17), Ensemble + OOD HE-CKA (Fig. 18), and Hypernetwork + OOD HE-CKA (Fig. 19). Each plot shows a grid comparing layerwise estimation of pairwise CKA, equipped with a linear feature kernel on the inlier Dirty-MNIST dataset, while using an unbiased estimator for CKA. To our surprise we found that $\text{CKA}_{\text{pw}}$ and HE-CKA results in fairly similar unbiased CKA estimates across the ensemble, but overall performance of the models in uncertainty estimation and accuracy presented in Table. 1 by HE-CKA were better. Nevertheless, methods utilizing $\text{CKA}_{\text{pw}}$ and HE-CKA kernels significantly reduce similarity of features compared to methods with no repulsive terms or RBF based kernels. This was especially true for our hypernetwork tests.

## G  Memory Footprint and Time Complexity

We compare the training runtime of our HE-CKA term to ParVI based methods presented in D' Angelo & Fortuin (2021). We evaluated mini-batch training time averaged over 50 batches on a Quadro RTX 8000. Each method used a ResNet18 fed with batches of 128 images from CIFAR-10. The results are presented in Table 8. Reported CUDA memory includes all ensemble members, loss, batch statistics, feature kernels (if applicable), and gradients. We see that all ParVI methods increase training time by 2.2x for 5 ensemble members and 1.2x for 10. Given that HE-CKA is applied layerwise, our method does require a slight increase in memory compared to the other ParVI methods, but is comparable to other methods in terms of training batch time increase. The time complexity for each minibatch of HE is $O(LN^2n^2)$ compared to typical function space $O(N^2n^2)$ or weight space $O(n^2)$ kernels, where $n$ is the number of particles, $N$ is the mini-batch size and $L$ is the number of layers we use to compute HE-CKA. However, $n$ and $N$ are typically small, 5 and up to 128 respectively in our experiments, and one can use a subset of the layers $L$, as we did with our experiments. While using the HE-CKA kernel does have an increased memory and computation cost than using the RBF kernel, the benefit of having a kernel invariant to feature permutations and scaling are worth the minor additional cost.

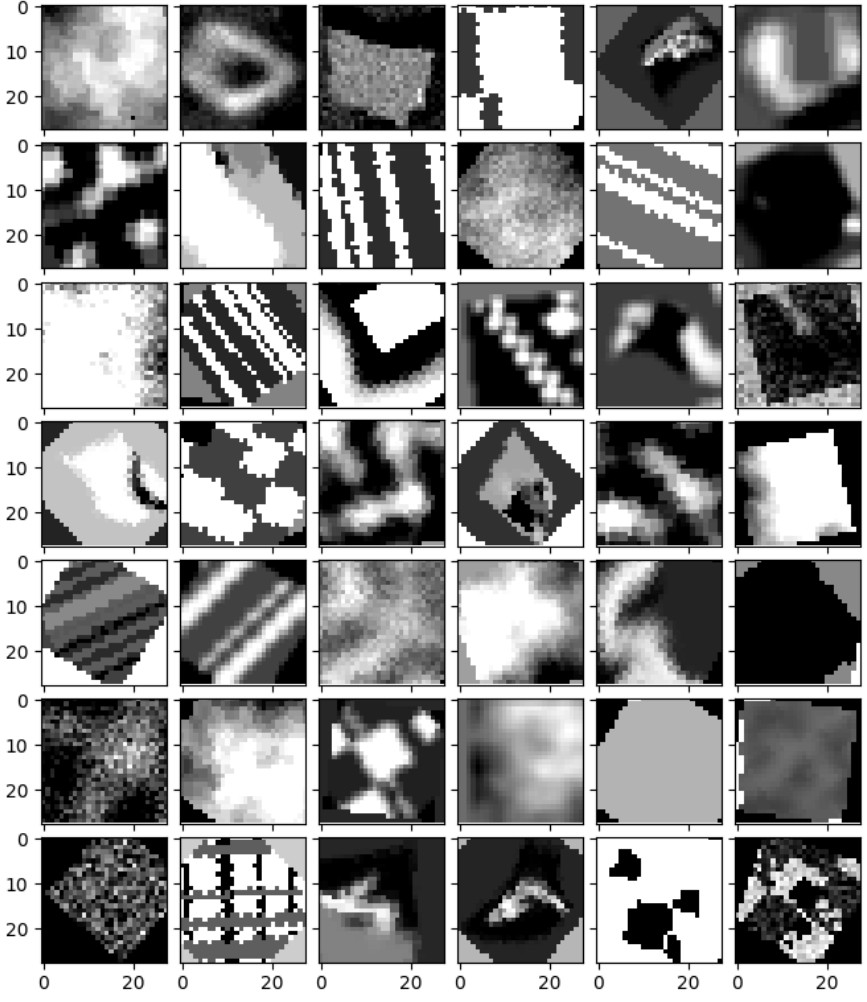

Figure 9: MNIST generated OOD set.

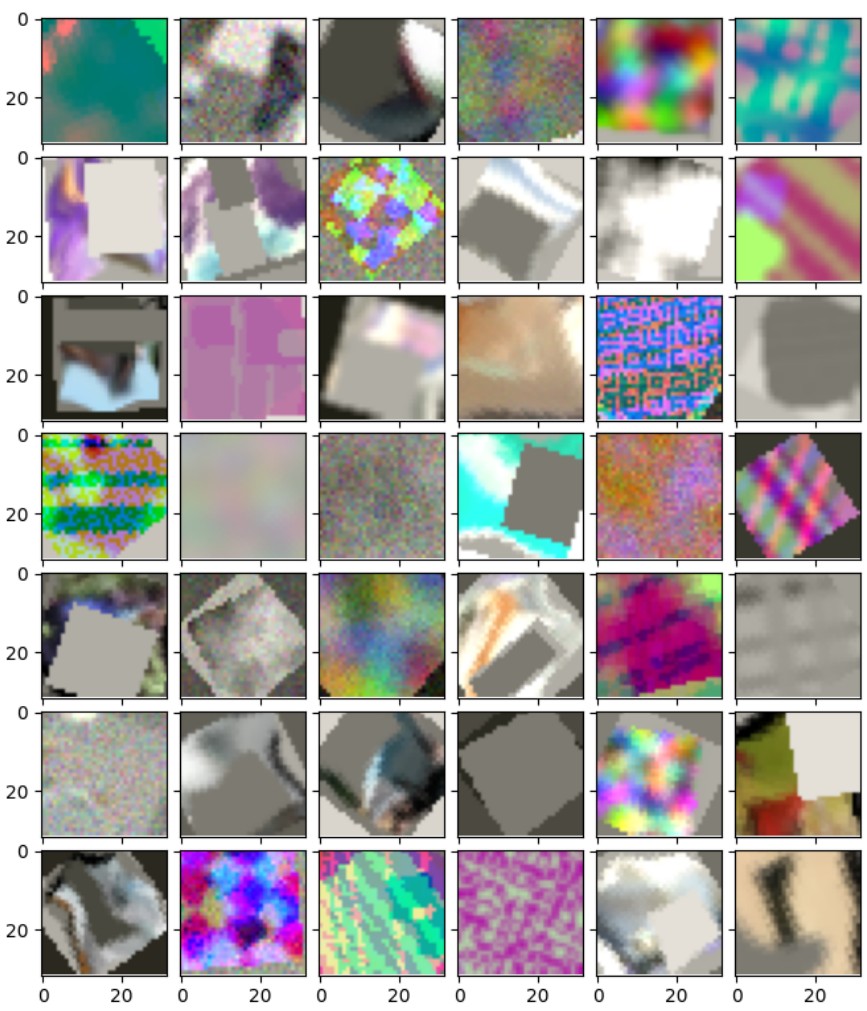

Figure 10: CIFAR generated OOD set.

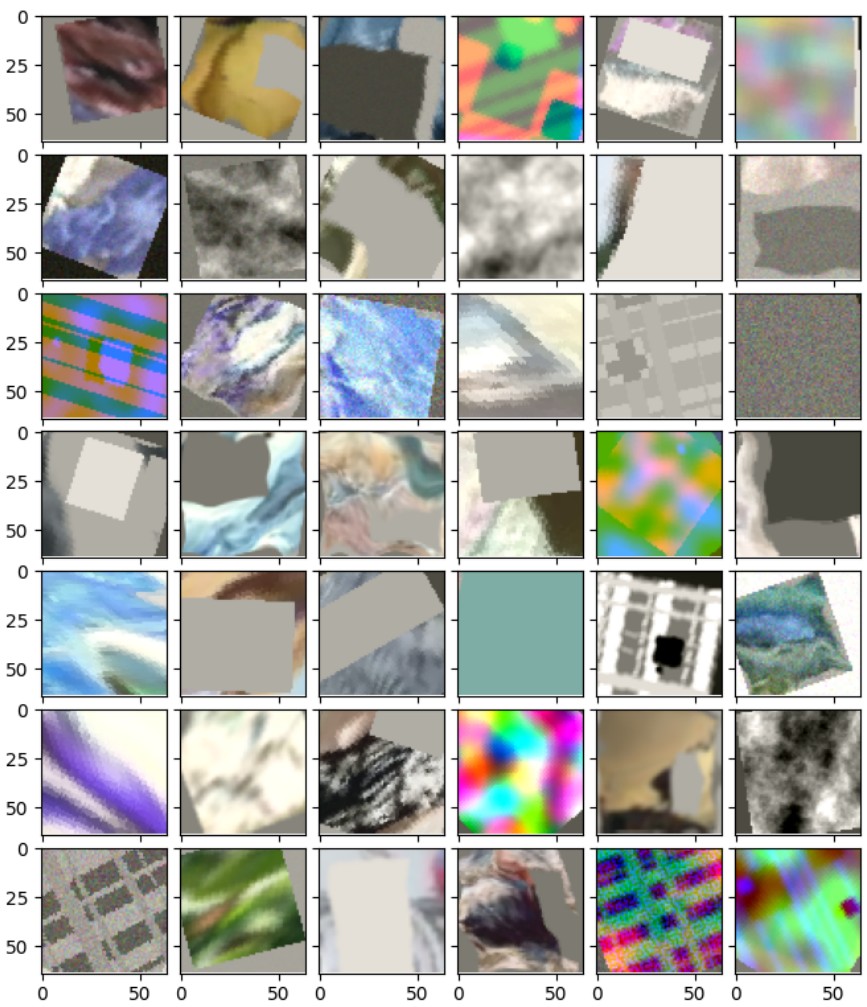

Figure 11: TinyImageNet generated OOD set.

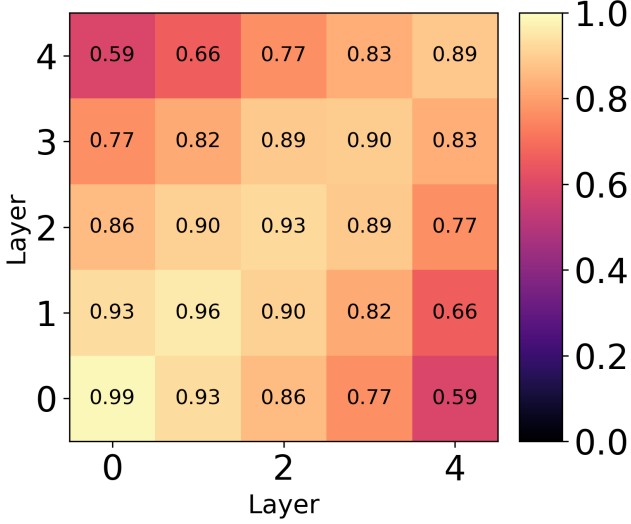

Figure 12: CKA values of a deep ensemble.

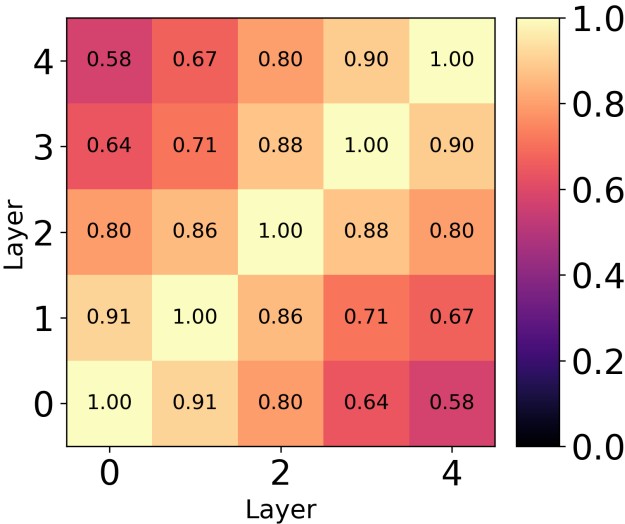

Figure 13: CKA values of a hypernetwork.

Table 8: Training compute and memory usage of a ensemble between various ParVI methods on CIFAR-10.

| Approach | Ensemble Size | CUDA Memory (GB) | Batch Time (ms) | Batch Time Increase |
|---|---|---|---|---|
| Deep Ensemble | 5 | 0.75 | $75 \pm 33$ | 1.00 |
| SVGD + RBF | 5 | 1.13 | $163 \pm 23$ | 2.17 |
| KDE-WGD + RBF | 5 | 1.14 | $165 \pm 17$ | 2.20 |
| SSGE-WGD + RBF | 5 | 1.58 | $194 \pm 22$ | 2.59 |
| HE-CKA (ours) | 5 | 1.65 | $163 \pm 34$ | 2.17 |
| Deep Ensemble | 10 | 1.35 | $297 \pm 36$ | 1.00 |
| SVGD + RBF | 10 | 2.18 | $346 \pm 25$ | 1.16 |
| KDE-WGD + RBF | 10 | 2.19 | $346 \pm 17$ | 1.16 |
| SSGE-WGD + RBF | 10 | 3.13 | $391 \pm 22$ | 1.32 |
| HE-CKA (ours) | 10 | 3.29 | $395 \pm 37$ | 1.33 |

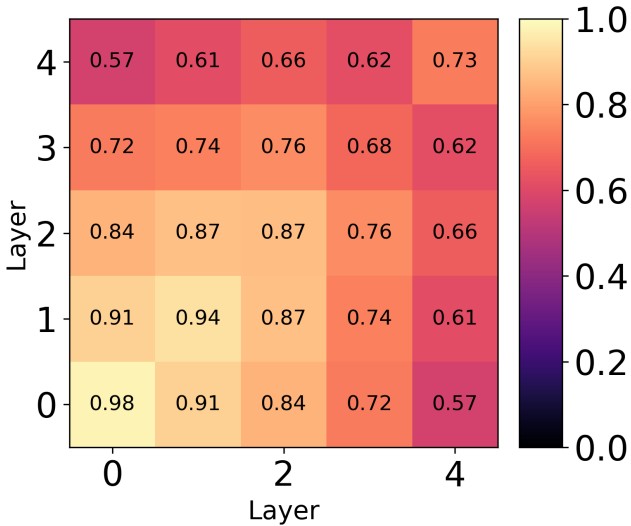

Figure 14: CKA values of an ensemble trained with SVGD + RBF.

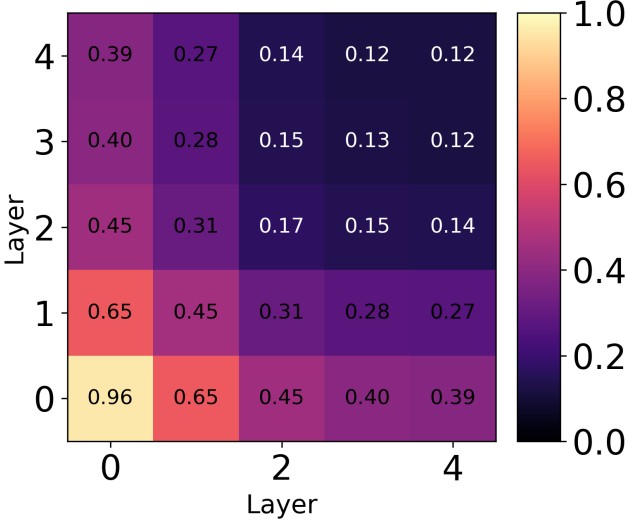

Figure 15: CKA values of a ensemble trained with $\text{CKA}_{\text{pw}}$ regularization.

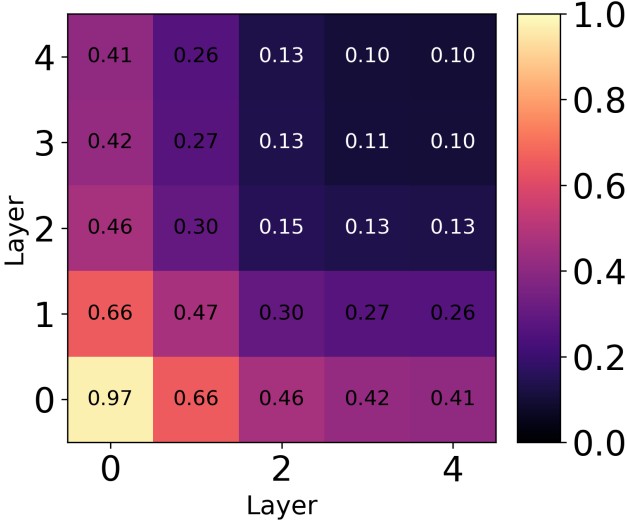

Figure 16: CKA values of a ensemble trained with SVGD + HE-CKA regularization.

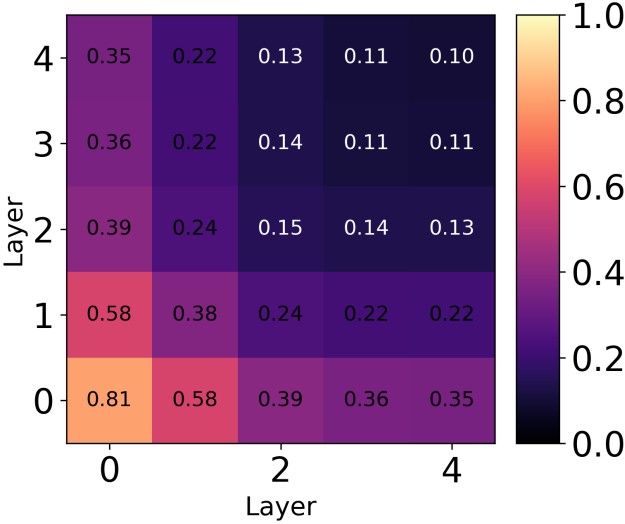

Figure 17: CKA values of a deep ensemble trained with HE-CKA regularization.

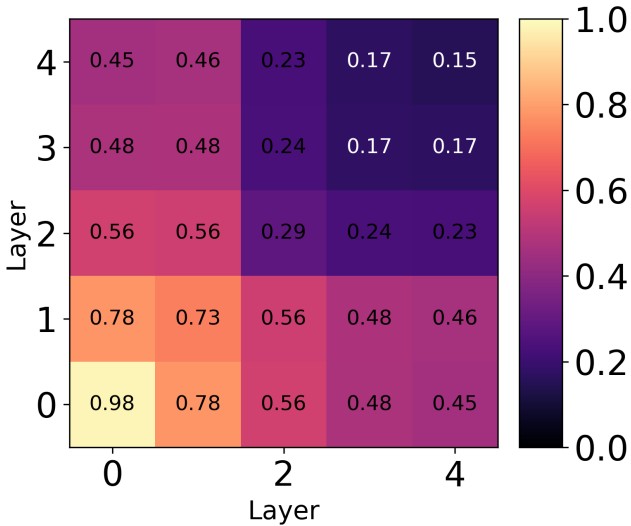

Figure 18: CKA values of a deep ensemble trained with OOD + HE-CKA and entropy terms.

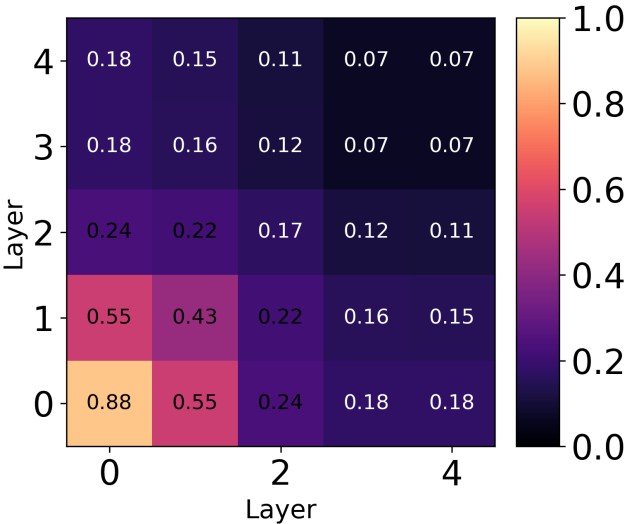

Figure 19: CKA values of a deep ensemble sampled from a hypernetwork trained with OOD HE-CKA and entropy terms .

