# OpenReview forum: "Enhancing Diversity in Bayesian Deep Learning via Hyperspherical Energy Minimization of CKA"
_NeurIPS.cc/2024/Conference — NeurIPS 2024 poster_

### Official Review · Reviewer_jUbb · 2024-07-10

**Soundness:** 2
**Presentation:** 3
**Contribution:** 2
**Rating:** 4
**Confidence:** 3

**Summary:**

This paper introduces the hyperspherical energy as an objective to prompt the diversity of particles in BNNs. It claims that the hyperspherical energy approach can avoid permutation invariance in traditional diversity metrics.

**Strengths:**

I agree that the diversity of particles in BNNs is an important problem. The proposed method introduces hyperspherical energy, which can avoid permutation invariance. This invariance is mainly due to the complex structures of DNNs, which previous Bayesian Inference methods have long ignored.

**Weaknesses:**

The technical contribution of this paper is not strong enough. Only a new regularization term is added to the standard ensemble framework. The idea of hyperspherical energy comes from previous works, and I do not recognize significant changes from previous works.

**Questions:**

1. The diversity evaluations are limited to synthetic data. I am not convinced that the proposed method can be scaled up.

2. Important ablation studies are missing. For example, different model architectures and the number of particles.

**Limitations:**

1. This paper is restricted to relatively small Bayesian neural networks, and does not discuss the scalability.

2. The proposed method can improve uncertainty estimation, but fails to improve accuracy.

---

> ### Author Rebuttal · Authors · 2024-08-07
>
> ## Technical Contribution
> We strongly disagree with the reviewer’s dismissal of our contribution as “merely a regularization term”. Comparing neural networks in particle-based variational inference (ParVI) has been a difficult problem and most proposed comparison kernels suffer from the lack of channel permutation invariance (i.e. just permuting channel positions lead the networks to be different from each other according to those kernels whereas the function they realize would be exactly the same), and our exploration of pairwise CKA kernels in ParVI solves this problem and provides a differentiable kernel that is invariant to channel permutations and isotropic scaling. These important properties have long been ignored in the ParVI. We additionally add HE to work on top of the CKA kernel, and our experiments demonstrated that it is more effective at minimizing a cosine similarity (in our case CKA) than minimizing cosine similarity itself (Section 3). We believe we are the first to adopt MHE for particle variational inference (ParVI) as well as the first to propose using MHE on top of a pairwise CKA kernel to compare networks. Other reviewers also believe that our approach is well-motivated/novel.
>
> ## Particle Number Ablation
> We agree that performing ablations on the number of particles is important. We observe improvements in inlier accuracy and outlier detection performance as the number of particles increases, with four and five particles performing similarly (Table 2). We will add more ablations on particle numbers in the final version of the paper.
>
>
> | Training Particles | NLL $\downarrow$ | ID Accuracy $\uparrow$ | ID ECE $\downarrow$ | AUROC SVHN $\uparrow$ | AUROC CIFAR-10/CIFAR-100 $\uparrow$ | AUROC Textures (DTD) $\uparrow$ |
> | -------- | -------- | -------- | -------- | -------- | -------- | -------- |
> | 2    | 0.791     | 59.21     | 5.21     | 89.87     | 67.34/68.44     |  68.48    |
> | 3    | 0.771     | 60.86     | 5.74     | 91.57     | 69.05/70.59     |  69.31    |
> | 4    | **0.761**     | 62.26     | 6.90     | **92.92**     | 70.83/71.44     |  **70.89**    |
> | 5   | 0.784     | **63.10**     |  9.82    | 92.65     | **72.13**/**71.68**     | 70.69 |
>
> **Table 2.** Ablation on number of training particles for ResNet18 + $\text{HE}$ trained with TinyImageNet.
>
> ## Questions
> Contrary to the reviewer’s claim, we did, evaluate on MNIST/CIFAR10/CIFAR100 in the original submission, and additionally TinyImageNet in the rebuttal (see general comments). None of these datasets are synthetic. Synthetic examples were included for visualization purposes but those weren’t the main results in the paper.
>
> ## Limitations
> We did discuss scaling and memory footprint in Appendix F. Results showed that our algorithm does not increase training time significantly over regular ensembles.

---

> > ### Comment · Reviewer_jUbb · 2024-08-08
> >
> > I acknowledge that the evaluations do cover MNIST/CIFAR10/CIFAR100. I am willing to raise my score to reflect that, along with the new experiments added in the rebuttal.
> >
> > However, I still do not think the technical contributions mentioned in the rebuttal are convincing. The rebuttal only discusses the meaning of this regulation term.

---

> ### Author Response · Authors · 2024-08-13
>
> We thank the reviewer for the recognition of our experimental results and the raised rating. We also appreciate the continuing discussion.
>
> We have discussed in our rebuttal the nontrivial changes and key properties of our proposed regularization method versus previous regularization techniques in ensemble training. As recognized by other reviewers, the combination of $\text{CKA}$ and $\text{HE}$ is a novel approach and a fresh perspective, and most importantly, it leads to improvements in practice for a long-standing important problem in ensemble learning, i.e., enabling better ensemble diversity (not attainable by merely switching channels within a network) which leads to significantly improved OOD performance.
>
> We disagree with the reviewer's attempt to dismiss any paper that works on regularization of networks to be not novel, without actually judging the contribution of the paper. Besides, our novel $\text{CKA}$+$\text{HE}$ kernel also allowed us to train for increased diversity on the synthetically generated outlier images which further significantly improved the OOD performance in all cases. To the best of our knowledge, this has not been explored in existing literature of ParVI methods.

---

### Official Review · Reviewer_Wp13 · 2024-07-12

**Soundness:** 3
**Presentation:** 3
**Contribution:** 3
**Rating:** 7
**Confidence:** 3

**Summary:**

The paper “Minimizing Hyperspherical Energy for Diverse Deep Ensembling” explores the use of Centered Kernel Alignment (CKA) and Minimization of Hyperspherical Energy (MHE) in Bayesian deep learning to enhance the diversity of ensemble models and parameters generated by hypernetworks. By incorporating these techniques, the authors aim to improve uncertainty quantification in both synthetic and real-world tasks, which is quantified by measuring OOD detection performance and calibration. The key contributions include proposing CKA as an optimization objective and utilizing MHE to address the diminishing gradient issue, leading to more stable training and better performance in uncertainty estimation.

**Strengths:**

- **Original:** The paper introduces a novel approach by combining CKA and MHE to enhance the diversity of deep learning ensembles and hypernetworks, which is a fresh perspective in Bayesian deep learning.
- **Detailed experiments:** The experimental results are comprehensive and demonstrate significant improvements in uncertainty quantification across various tasks, showing the practical effectiveness of the proposed methods.
- **Well-written:** The paper is well-structured, with clear explanations of the methods and thorough discussions of the results. The figures and tables effectively illustrate the performance improvements.

**Weaknesses:**

- The comparison of the approach in terms of OOD detection performance is slightly biased due to leveraging of generated OOD samples. Here in order to differentiate the contribution of the method from the incorporation of additional information a further experiments comparing against DDU would be warranted. In particular one could imagine adding a term to the loss which encourages low likelihood of the fitted GMM on OOD samples.
 - The comparisons in Figure 2 paint a slightly overly optimistic picture for the OOD HE approaches. In 2 dimensions the problem becomes quite trivial given negative / OOD samples. The difficulty of generating enough samples to cover the OOD volume becomes exponentially more difficult with increasing dimensionality of the input space and is especially easy when the diversity of the data is limited.
 - (Related to prior point) the datasets used to show the efficacy of the OOD approach are rather simple and small such that sample diversity is limited. Running one experiment on a larger dataset like imagenet would be very beneficial to show that the method can also work in the context of highly diverse input distributions and high dimensionality of the input space.
 - There seems to be a mistake in table 2 row Ensemble + HE, column PE
 - I recommend renaming Fig1 d and e to Cossim evaluation and HE evaluation as using objective gives the impression that the data shown is from models trained using these objectives.
 - As the work points out that applying the method to larger datasets would in principle be possible, this might be a good additional comparison to include. Many comparison approaches will have issues in this case, but comparison to other ensembling based approaches would be feasible.

**Questions:**

- When running the LeNet comparisons agains DDU, was the model trained with spectral normalization (a requirement for DDU to function well) and how was the regularization strength determined?
- Why was the normalization-free variant of LeNet used for some methods whereas the standard variant for the DDU baseline?
- Page 9, line 289: Was the GMM fitted to the features of the pre-classification layer. I.e. to the features used by the last linear layer?

**Limitations:**

The work correctly points out that the small size of the datasets is one of the main limitations. Generally, seeing the efficacy of the approach in a larger scale training setup would significantly improve the paper.

---

> ### Author Rebuttal · Authors · 2024-08-07
>
> ## Questions
> >When running the LeNet comparisons against DDU, was the model trained with spectral normalization (a requirement for DDU to function well) and how was the regularization strength determined?
>
> Yes, all DDU models were trained with spectral normalization. We picked the best regularization for experiment balancing AUROC/inlier accuracy from values tested in the range $1.0-5.0$.
>
> >Why was the normalization-free variant of LeNet used for some methods whereas the standard variant for the DDU baseline?
>
> To ease the transition to hypernetworks, we aimed to construct a method not requiring feature normalization. We utilized weight standardization (WS) but did not adapt WS to work along with spectral normalization, and found no difference in inlier performance.
>
> >Page 9, line 289: Was the GMM fitted to the features of the pre-classification layer. I.e. to the features used by the last linear layer?
>
> Yes, the GMM was fitted to the features of the pre-classification layer.

---

> > ### Comment · Reviewer_Wp13 · 2024-08-12
> >
> > Thanks for answering the questions. I would have appreciated some comments on the weaknesses of the work I pointed out. Will leave my score unchanged as solely the issue of dataset size was addressed and other points remained unmentioned.

---

> > > ### Author Response · Authors · 2024-08-13
> > >
> > > Thank you for pointing out the missing response to your comments on the weaknesses. We somehow forgot to include them and will post the answers here in addition to the experiments on the larger dataset.
> > >
> > > > Fair comparison against DDU.
> > >
> > > Instead of adding OOD examples to DDU as you suggested, which is also viable, in Tables (Paper 1-4. Rebuttal Table 1), we showed that even without OOD samples, our method/kernel still outperforms ensembles and DDU (Tables 1/2) in OOD detection, which are the apples-to-apples comparisons. Although, we can mark that row more clearly to indicate synthetic OOD sample usage.
> > >
> > > > In 2D with OOD examples the problem becomes quite trivial. The difficulty of generating enough samples to cover the OOD volume becomes exponentially more difficult with increasing dimensionality of the input space and is especially easy when the diversity of the data is limited.
> > >
> > > We agree that adding OOD to synthetic experiments is easy, however as mentioned earlier our methods outperform baselines even without OOD examples. For images, our main point is to show that we do not have to cover the entire OOD volume but just have to generate some random images to make sure the diversity is high on random outlier images. Our OOD detection experiments show that although our generated synthetic images (Fig. 7/8) are nowhere similar to the tested outliers, our approach still performs well for outlier detection.
> > >
> > > > Larger datasets
> > >
> > > This one is answered in the general remarks.
> > >
> > > > Table 2 and Figure 1:
> > >
> > > Thank you for pointing these out. We will correct the extra zero in the table, and further clarify the cossim/he figure name.

---

### Official Review · Reviewer_JKj8 · 2024-07-14

**Soundness:** 3
**Presentation:** 3
**Contribution:** 3
**Rating:** 6
**Confidence:** 4

**Summary:**

The authors proposed to improve the quantification of particle diversity in deep ensemble with hyperspherical energy (HE) on top of the CKA kernel. They further integrate the HE kernel in particle-based variational inference (ParVI) and generative ensemble with hypernetwork frameworks. The methods are evaluated on both synthetic experiments and small-scale classification datasets.

**Strengths:**

1. The motivation of the paper is clear: addressing mode collapse in deep ensemble by using a kernel that is more suitable to measure particle diversity of neural networks (HE kernel).
2. The advantage of HE kernel has been demonstrated in two different ensemble frameworks (parVI & generative ensemble with hypernetwork), and the empiricall performance of the method looks Ok.

**Weaknesses:**

The datasets considered are a bit outdated and the networks considred seem to be quite small, such that overall the performance is on the lower end (e.g. 85% acc. for CIFAR10, while many BDL methods can easily achieve accuracy higher than 90%). Fuerthermore, recent BDL papers typically consider larger datasets (such as ImageNet) and deeper networks. It is necessary to consider larger datasets and larger models in order to assess the practical effectiveness of the method.

**Questions:**

See weakness

**Limitations:**

The authors do not address potential negative social impact since the paper is predominantly theoretical.

---

> ### Author Rebuttal · Authors · 2024-08-07
>
> ### CIFAR-10 Performance
> We did evaluate our approach using a standard ResNet18, shown in Table 4 of the manuscript. The inlier performance of all approaches (The “Accuracy” column in Table 4.) is above 95%. We compared using the permutation invariant kernels $\text{CKA}_\text{pw}$, $\text{HE}$ with RBF. Further explanation can be found in the general remarks.

---

> > ### Author Response · Authors · 2024-08-08
> >
> > For the question you asked about larger dataset and larger models, results can also be found in the general remarks.

---

> > ### Comment · Reviewer_JKj8 · 2024-08-12
> >
> > Thanks for the rebuttal, which addresses my concern about the model perfoemance. I am willing to raise my score to 6.

---

### Author Rebuttal · Authors · 2024-08-07

## General Remarks
We are grateful to the reviewers for their constructive feedback. Below, we respond to questions and concerns shared by the reviewers regarding running on a larger dataset.


## Larger Dataset
We agree with the reviewers that the evaluation of BDL on a larger dataset is desirable. Due to time constraints of the rebuttal period, we applied HE to an ensemble of ResNet18 on TinyImageNet (Table 1). Encouraging feature diversity with $\text{CKA}_\text{pw}$ and $\text{HE}$ significantly improves uncertainty estimation while minimally impacting inlier performance. Besides, encouraging feature diversity on synthetically generated OOD examples, albeit highly different from the testing examples, further significantly improves the performance of both inlier classification and outlier detection. We will add more ImageNet-level experiments in the final version.

| Model | NLL $\downarrow$ | ID Accuracy $\uparrow$ | ID ECE $\downarrow$ | AUROC SVHN $\uparrow$ | AUROC CIFAR-10/CIFAR-100 $\uparrow$ | AUROC Textures (DTD) $\uparrow$ |
| -------- | -------- | -------- | -------- | -------- | -------- | -------- |
| ResNet18 (5)    | 0.775     | 62.95     | 8.90     | 89.81     | 66.85/67.33     |  68.96    |
| SVGD + RBF (5)     | 0.926     | 61.87     |  16.10    | 92.76     | 72.23/73.73     | 65.67     |
| SVGD + $\text{CKA}_\text{pw}$ (5)     | 0.835     | 60.15     |  8.26    | 94.08     | 78.40/79.48     | 66.48     |
| SVGD + $\text{HE}$ (5) | 0.732 | 61.36 | **3.71** | 94.10 | 72.05/72.86 | 70.75 |
| ResNet18 + $\text{HE}$ (5)     | 0.784     | 63.10     |  9.82    | 92.65     | 72.13/71.68     | 70.69 |
| ResNet18 + $\text{HE}$ OOD (5)   | **0.606**     | **68.51**     | 11.49     | **98.54**     | **79.00**/**81.09**     | **89.18** |

**Table 1.** Performance of ResNet18 ensemble trained with TinyImageNet. All models are pretrained with a deep ensemble with no regularization, then fine tuned for 20 epochs with each method (including the deep ensemble). Standard predicted entropy is used in the AUROC calculation. Synthetic OOD generated from noise and augmented TinyImagenet.


## Larger Models
We have already included results with a ResNet18 with over **96%** inlier accuracy performance on CIFAR-10 in the original manuscript (Table 4 (Column 4 Accuracy is inlier accuracy)). The smaller ResNet32 models,  with ~85% inlier performance (Table 2.) were additionally included for a fair one-to-one comparison with the work presented in (D’Angelo et al. 2021), . Additionally, we reported scaling time/memory costs with typical weight space function space ParVI methods in Appendix F. Our approach is comparable to other ParVI methods in speed and incurs a slightly larger memory cost due to feature kernel construction.

---

> ### Author Response · Authors · 2024-08-14
>
> ## TinyImageNet Correction
> We discovered a bug in our TinyImageNet dataloader for the HE + OOD experiment during the rebuttal period, which skewed our results for that one experiment. After fixing it and retraining, we obtained the following updated results.
>
> | Model | NLL $\downarrow$ | ID Accuracy $\uparrow$ | ID ECE $\downarrow$ | AUROC SVHN $\uparrow$ | AUROC CIFAR-10/CIFAR-100 $\uparrow$ | AUROC Textures (DTD) $\uparrow$ |
> | -------- | -------- | -------- | -------- | -------- | -------- | -------- |
> | ResNet18 + $\text{HE}$ OOD (5)   | 0.786     | 61.88     | 8.02     | **99.31**     | **81.56**/**87.64**     | **90.94** |

---

### Decision · Program_Chairs · 2024-09-25

**Decision:**

Accept (poster)

**Comment:**

In this work, the authors propose a novel approach to increase feature diversity in deep ensembles by minimizing hyperspherical energy (MHE) on top of Centered Kernel Alignment (CKA) kernels. The reviewers praise the work for its originality, well-written presentation, and comprehensive experiments demonstrating significant improvements in uncertainty quantification. However, they are critical of the limited evaluation on small datasets and the lack of ablation studies on particle numbers and model architectures. In their rebuttal, the authors clarify their contributions, address concerns about dataset size and model performance, and provide additional experiments on TinyImageNet. This addressed the reviewers' concerns about dataset size and model performance but did not fully address their concerns about the technical contribution and scalability.

After the rebuttal and discussion, the paper is very borderline, where two reviewers lean towards acceptance, while one leans towards rejection. The main criticism on which the decision hinges seems to be the small scale of the experiments and the lack of ablation studies. Both of these have been provided by the authors in their rebuttal.  I trust that the authors add those additional results to the camera-ready version and might even add more ablation studies that have not yet been provided, which would further strengthen the paper. Based on this, I recommend acceptance. However, I would further recommend that the authors seriously take all the reviewer feedback into account for the camera-ready version.